# Self-Supervised Learning as Discrete Communication

**Kawtar Zaher** [* 1 2]   **Ilyass Moummad** [* 1]   **Olivier Buisson** [2]   **Alexis Joly** [1]

## Abstract

Most self-supervised learning (SSL) methods learn continuous visual representations by aligning different views of the same input, offering limited control over how information is structured across representation dimensions. In this work, we frame visual self-supervised learning as a discrete communication process between a teacher and a student network, where semantic information is transmitted through a fixed-capacity binary channel. Rather than aligning continuous features, the student predicts multi-label binary messages produced by the teacher. Discrete agreement is enforced through an element-wise binary cross-entropy objective, while a coding-rate regularization term encourages effective utilization of the constrained channel, promoting structured representations. We further show that periodically reinitializing the projection head strengthens this effect by encouraging embeddings that remain predictive across multiple discrete encodings. Extensive experiments demonstrate consistent improvements over continuous agreement baselines on image classification, retrieval, and dense visual prediction tasks, as well as under domain shift through self-supervised adaptation. Beyond backbone representations, we analyze the learned binary codes and show that they form a compact and informative discrete language, capturing semantic factors reusable across classes.

## 1. Introduction

Self-supervised learning (SSL) has emerged as the predominant paradigm for learning visual representations without human annotations. Contemporary SSL methods can be broadly categorized into representation alignment approaches, which enforce consistency between different views of the same image through contrastive or self-distillation objectives (Chen et al., 2020; He et al., 2020; Caron et al., 2021; Bardes et al., 2021; Zhou et al., 2021; Oquab et al., 2023; Wu et al., 2025), and generative or predictive approaches, which learn representations by reconstructing or predicting missing content in pixel space (He et al., 2022; Yang et al., 2025), token space (El-Nouby et al., 2024), or embedding space (Assran et al., 2023; Xu et al., 2025). Despite their methodological differences, most of these frameworks rely on continuous-valued targets and similarity- or regression-based objectives, such as cross-entropy over prototypes or cosine distance minimization. While highly effective, continuous alignment enforces semantic consistency through global similarity in the embedding space, offering limited explicit control over representational structure. This may result in entangled features with multiple semantic factors mixed across dimensions.

Discrete representations provide a compelling alternative to address these limitations. Encoding information through binary variables allows each dimension to capture the presence or absence of a semantic attribute, naturally encouraging a decomposition of information across dimensions and enabling multi-label semantics. Such representations are well suited to capture the compositional structure of visual scenes and, in principle, can support up to $2^d$ distinct configurations in $d$ dimensions. However, in self-supervised learning, discrete representations have primarily been explored in reconstruction-based settings, using fixed codebooks for masked image modeling (Bao et al., 2021; Dong et al., 2023). These approaches are often outperformed by representation alignment methods on real-world data (Van Assel et al., 2025), and several recent works report that continuous targets remain more effective than discrete token-based formulations in predictive SSL objectives (He et al., 2022; El-Nouby et al., 2024; Xu et al., 2025). In supervised settings, discrete multi-label representations can be defined using ground-truth annotations (Lee et al., 2022), but without labels it remains unclear how to define informative discrete targets or ensure that each dimension carries meaningful distinct information.

In this work, we show that discrete representations can

[*]Equal contribution   [1]INRIA, LIRMM, Université de Montpellier, Montpellier, France   [2]Institut National de l'Audiovisuel, Paris, France. Correspondence to: Kawtar Zaher <kawtarzaher1999@gmail.com>, Ilyass Moummad <ilyass.moummad@inria.fr>, Alexis Joly <alexis.joly@inria.fr>.

*Proceedings of the 43rd International Conference on Machine Learning*, Seoul, South Korea. PMLR 306, 2026. Copyright 2026 by the author(s).

be effectively leveraged for self-supervised representation alignment. We propose to formulate SSL as a discrete communication problem between a teacher and a student, where information is transmitted through a fixed number of binary channels. The student predicts multi-label binary signals produced by the teacher using an element-wise binary cross-entropy objective, while a coding-rate regularization term encourages effective and balanced utilization of the discrete bottleneck. Unlike prototype-based SSL methods (YM. et al., 2020; Caron et al., 2020; 2021; Assran et al., 2022; Oquab et al., 2023; Siméoni et al., 2025), which enforce agreement around a single dominant prototype per image, our framework allows multiple semantic factors to be active simultaneously, yielding structured, factorized, and reusable representations. The discrete communication task serves solely as a self-supervised pretext objective: the backbone continues to learn continuous visual representations.

We evaluate the proposed discrete communication framework through visual representation learning experiments across a range of downstream tasks. Our approach matches or improves ImageNet-1K classification performance and yields consistent gains in image retrieval on in-domain and lightly out-of-distribution benchmarks, while also improving performance on unsupervised object detection, instance segmentation, and semi-supervised video object segmentation. Under severe domain shift, our model also achieves higher linear probing performance, which can be further improved through continued self-supervised fine-tuning on the target domain. Finally, we analyze both the learned continuous embeddings and the binary codes, showing that discrete agreement leads to more balanced and less correlated continuous representations, and yields compact and informative binary codes that capture reusable semantic factors shared across classes.

## 2. Related Work

**Self-Supervised Learning and Representation Alignment.** Most successful self-supervised learning (SSL) approaches rely on representation alignment across augmented views, using contrastive objectives (Chen et al., 2020; Tian et al., 2020; He et al., 2020) or teacher–student self-distillation (Grill et al., 2020; Caron et al., 2021; Zhou et al., 2021; Wu et al., 2025) to enforce invariance. In teacher–student frameworks, the student predicts momentum-updated teacher representations using similarity or regression losses in continuous embedding spaces, without requiring explicit negative samples (Grill et al., 2020; Caron et al., 2021; Wu et al., 2025). In parallel, reconstruction-based methods learn representations by recovering masked pixels or tokens (Bao et al., 2021; Dong et al., 2023; He et al., 2022; El-Nouby et al., 2024; Xu et al., 2025), but generally underperform alignment-based

objectives for learning general-purpose representations on real-world data (Van Assel et al., 2025). While approaches such as BEiT (Bao et al., 2021) introduce discrete prediction targets through a fixed tokenizer, the discretization applies to the reconstruction objective rather than to the agreement mechanism or the structure of the learned representation.

Our work builds on alignment-based self-supervised learning, but departs from prior approaches by replacing continuous alignment with explicit multi-label discrete agreement, enabling direct control over the structure and capacity of the learned representations. While alignment-based approaches such as SimCLR (Chen et al., 2020), BYOL (Grill et al., 2020), VICReg (Bardes et al., 2021), and SimDINO (Wu et al., 2025) rely on continuous similarity or regression objectives, and methods such as DINO (Caron et al., 2021) or iBOT (Zhou et al., 2021) employ soft assignment to prototypes, the supervision signals they induce remain continuous and predominantly mono-modal. In contrast, our approach introduces an explicit discrete agreement mechanism based on multi-label binary signals, while still learning continuous representations in the backbone. This distinction allows us to impose structured constraints on how information is allocated across dimensions of the agreement space, without discretizing the learned embeddings themselves.

**Information-Theoretic Perspectives on Self-Supervised Learning.** A growing body of work has analyzed self-supervised learning through an information-theoretic lens, viewing representation learning as the problem of maximizing informative content while avoiding collapse. Early formulations connect SSL objectives to mutual information maximization between augmented views (Ozsoy et al., 2022; Hjelm et al., 2018), typically approximated through contrastive or similarity-based losses. More recent approaches emphasize coding-rate and rate–distortion principles to characterize representation diversity and compression (Li et al., 2022b; Tong et al., 2023; Wu et al., 2025). In particular, the work of Yu *et al.* (Yu et al., 2020b) introduces the Maximal Coding Rate Reduction (MCR$^2$) principle, motivating objectives that balance alignment with expressiveness (Yu et al., 2020a; Tong et al., 2023; Wu et al., 2025). SimDINO (Wu et al., 2025) further connects self-distillation methods in the DINO family to coding-rate maximization as an effective anti-collapse mechanism. Our work is closely related to these perspectives, but differs in how coding-rate principles are operationalized. Rather than using coding rate implicitly to regularize continuous embeddings, we introduce an explicit multi-label discrete bottleneck and encourage its effective utilization through coding-rate regularization. This combination constrains the total information capacity while promoting its balanced allocation across dimensions, which in turn encourages factorized representations. As a result, the learned representations can support semantic attributes that are shared across images and classes, rather than being

tied to mutually exclusive categories.

**Hashing.** Deep hashing methods aim to learn compact binary codes for efficient similarity search, typically framing code learning as a compression problem and distilling knowledge from pretrained encoders to preserve neighborhood structure in Hamming space (Luo et al., 2023). Recent self-supervised hashing approaches incorporate contrastive, reconstruction-based, or regularization objectives to mitigate code collapse and improve retrieval performance (Cao et al., 2023; Ma et al., 2024; Gong et al., 2022; Shen et al., 2024; Li & van Gemert, 2021; Moummad et al., 2025). In contrast to these works, our objective is not to learn binary codes as final representations. Instead, we train teacher–student models from scratch using a hashing-like objective as a self-supervised pretext task, where binary codes serve as discrete prediction targets. Importantly, while binary cross-entropy losses are commonly used in hashing to optimize similarity-preserving codes, they are typically embedded in pairwise or neighborhood-based formulations, whereas our loss enforces multi-label discrete agreement between teacher and student and does not directly optimize the codes themselves for retrieval.

## 3. Problem Setup

We consider two random variables $X_1, X_2$ representing two stochastic augmentations of the same image $I \in \mathcal{X}$:

$$X_1 = A_1(I), \qquad X_2 = A_2(I), \qquad (1)$$

where $A_1, A_2$ are random augmentation operators (crop, color jitter, blur, *etc.*), and $\mathcal{X}$ denotes the space of images.

We aim to learn a representation mapping $f_\theta : \mathcal{X} \to \mathbb{R}^d$ and a *binary communication head* $\tilde{g}_\phi$ that parameterizes a discrete representation space of fixed capacity. Specifically, $\tilde{g}_\phi$ maps continuous features to a factorized Bernoulli distribution over $B$ binary variables:

$$\tilde{g}_\phi : \mathbb{R}^d \to (0,1)^B. \qquad (2)$$

Given an input $x$ (either $X_1$ or $X_2$), the encoder produces a feature $h = f_\theta(x) \in \mathbb{R}^d$, and the head outputs Bernoulli parameters $p = \tilde{g}_\phi(h)$. A binary code $z \in \{0,1\}^B$ is then drawn as:

$$z \sim P_\phi(Z \mid X) = \prod_{b=1}^{B} \text{Bernoulli}(p_b). \qquad (3)$$

In the following, we use this probabilistic formulation to define our objective, while practical optimization relies on tractable surrogates introduced in the next section.

We seek binary codes $Z_1$ and $Z_2$ associated with $X_1$ and $X_2$ that are:

(i) invariant across augmentations,

(ii) spread out across samples and approximately factorized across bits.

From a communication perspective, $Z_1$ and $Z_2$ can be viewed as two messages transmitted through the same discrete channel under different noise realizations induced by stochastic data augmentations. From this perspective, our learning objective can be naturally formulated in information-theoretic terms:

$$\max_{\theta,\phi} \; \mathbb{E}\big[I(Z_1; Z_2)\big] \qquad (4)$$

$$\text{where} \qquad I(Z_1; Z_2) = H(Z_1) - H(Z_1|Z_2).$$

Thus, maximizing the mutual information encourages invariance through the conditional entropy minimization, while the entropy maximization enforces maximal utilization of the $B$-bit discrete communication channel, leading to diverse and approximately independent bits.

The $B$-bit bound on channel capacity is a direct consequence of our probabilistic formulation. As each binary variable $z_b \in \{0,1\}$ follows a Bernoulli distribution, the communicated $B$-dimensional binary message $Z$ has theoretically bounded entropy: $H(Z) \leq B$. Consequently, the mutual information between the two views is also bounded:

$$I(Z_1; Z_2) \leq B. \qquad (5)$$

This enforces a fixed-capacity communication channel between the two codes. Importantly, this constraint applies to the communication channel, not to the learned representations $h$, which remain continuous.

This is a fundamental difference from the numerical precision of standard continuous agreement objectives. In such settings, each component of the communicated vector can be represented with a *float32* variable, and can represent up to $2^{32}$ different values. However, its entropy is not controlled by the setup, and mainly depends on the learned distribution (e.g., $H = 32$ if uniform, $H = 1$ if restricted to 0/1). In contrast, our binary formulation makes the maximal channel capacity an explicit architectural constant, fixed independently of the data or the training objective. As a consequence, redundancy across bits directly wastes available capacity: if two bits encode the same information, the effective channel capacity drops below $B$, making the objective suboptimal. This induces a pressure to distribute information across dimensions, leading to the underlying information factorization, later observed empirically.

In practice, we use $\tilde{g}_\phi = \sigma \circ g_\phi$ where $g_\phi : \mathbb{R}^d \to \mathbb{R}^B$ is a continuous projection head, and $\sigma$ denotes the element-wise sigmoid function.

In the next section, we show how each component of the objective in Eq. 4 is approximated and optimized in practice.

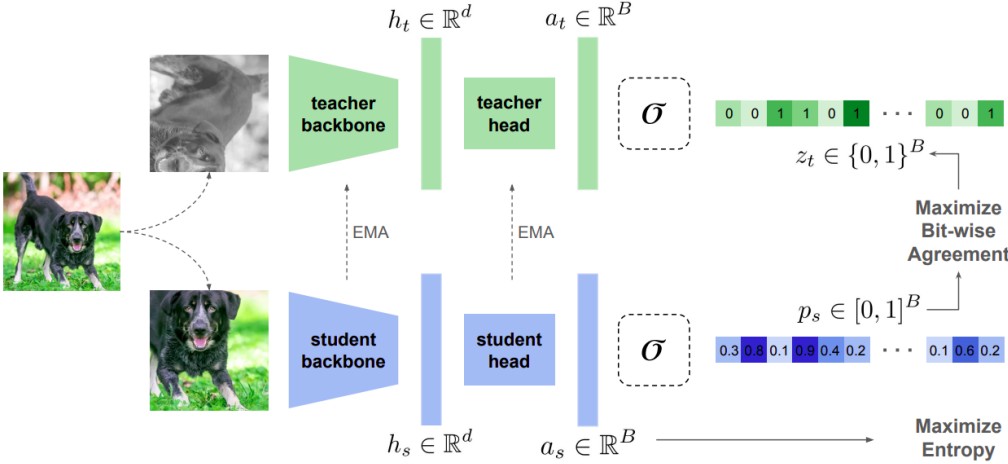

*Figure 1.* Overview of our self-supervised learning as discrete communication pipeline: Two augmented views are processed by teacher and student networks. The teacher outputs are binarized by thresholding sigmoid $\sigma$ probabilities at 0.5 to produce binary targets, while the student is trained via backpropagation to predict them through a binary cross-entropy objective (alignment). A coding-rate regularization is applied, as an entropy maximization proxy, to the student logits $a_s$ to encourage diverse and decorrelated representations (anti-collapse regularization). The teacher parameters are updated using EMA of the student parameters.

## 4. Method

### 4.1. Architecture

We build on the simplified DINO framework proposed in SimDINO (Wu et al., 2025), which removes prototype assignments and contrastive normalization, resulting in a cleaner alignment objective that better isolates the effects of our discrete agreement mechanism. Our pipeline is illustrated in Fig. 1. Similar to their work, our model consists of:

- a backbone $f_\theta : \mathcal{X} \to \mathbb{R}^d$ (e.g., a Vision Transformer),
- a projection head $g_\phi : \mathbb{R}^d \to \mathbb{R}^B$, typically a small MLP.

For each training step, we sample two augmentations $x_1 \sim X_1$ and $x_2 \sim X_2$ of the same image $I$, and we compute the embeddings $h_i$, apply the projection head to obtain the logits $a_i$ and the probabilities $p_i$, where $i \in \{1, 2\}$:

$$h_i = f_\theta(x_i), \quad a_i = g_\phi(h_i), \quad p_i = \sigma(a_i). \quad (6)$$

In our discrete communication perspective, the projection head $g_\phi$ parameterizes the discrete communication channel, while the backbone $f_\theta$ learns representations that are robust to noise induced by data augmentations.

### 4.2. Binary Agreement Objective

The invariance term (i) of Eq. 4 corresponds to minimizing the conditional entropy $H(Z_1|Z_2)$, which enforces consistency between the binary codes associated with different augmentations of the same image.

As the true conditional distribution $P(Z_1|Z_2)$ is unknown and intractable, we introduce a variational surrogate distribution $Q_\phi(Z_1|Z_2)$, which yields the standard variational upper bound:

$$H(Z_1|Z_2) \leq \mathbb{E}\big[ -log(Q_\phi(Z_1|Z_2)\big],$$

thus: $\max\ I(Z_1; Z_2) \iff \min\ \mathbb{E}\big[ -log(Q_\phi(Z_1|Z_2)\big].$

The framework consists of two branches: a teacher branch $(f_\theta^t, g_\phi^t)$, and a student branch $(f_\theta^s, g_\phi^s)$. Since $Z_2$ is induced from $X_2$, we model $Q_\phi(Z_1|Z_2)$ as the factorized Bernoulli distribution whose parameters are predicted by the student network applied on $X_2$:

$$Q_\phi(Z_1 \mid Z_2) = \prod_{b=1}^{B} \text{Bernoulli}(p_{2,b}^s). \quad (7)$$

We define the binary targets by applying a stop-gradient threshold to the probabilities obtained via the teacher branch:

$$\hat{z}_1^t = \mathbf{1}[p_1^t > 1/2], \qquad \hat{z}_2^t = \mathbf{1}[p_2^t > 1/2]. \quad (8)$$

We use hard thresholded targets to provide an explicit and unambiguous discrete supervision signal, while gradients are propagated only through the student branch.

Assuming conditional independence across bits, the negative log-likelihood of the teacher code $\hat{z}_1^t$ under $Q_\phi(Z_1|Z_2)$ reduces to an element-wise binary cross-entropy. We use a

symmetric cross-view formulation:

$$\mathcal{L}_{BCE} = -\sum_{b=1}^{B} \left( \hat{z}_{1,b}^t \log p_{2,b}^s + (1 - \hat{z}_{1,b}^t) \log(1 - p_{2,b}^s) \right.$$
$$\left. + \hat{z}_{2,b}^t \log p_{1,b}^s + (1 - \hat{z}_{2,b}^t) \log(1 - p_{1,b}^s) \right). \tag{9}$$

This loss encourages the student to accurately recover the discrete message communicated by the teacher, implementing a bit-wise approximation of the conditional entropy minimization in Eq. 4.

### 4.3. Coding-Rate Regularization

The binary agreement objective introduced above provides a practical approximation of the conditional entropy term $H(Z_1 \mid Z_2)$ in Eq. 4, enforcing invariance across augmentations under a fixed-capacity discrete bottleneck. However, maximizing mutual information also requires maximizing the marginal entropy $H(Z)$, which controls how effectively the available capacity of the discrete channel is utilized.

From an information-theoretic perspective, maximizing $H(Z)$ for a fixed-length binary code amounts to promoting high marginal entropy and low redundancy across bits. In practice, this corresponds to encouraging representations that are diverse across samples and approximately uncorrelated across dimensions, so that each bit contributes complementary information. While these objectives are naturally defined over the discrete variables, directly operating on hard-thresholded codes is challenging both due to non-differentiability and because estimating marginal or joint entropies over high-dimensional binary codes is inherently combinatorial.

For this reason, we regularize the continuous pre-binarization logits rather than the discrete codes themselves. While SimDINO applies a coding-rate regularization to L2-normalized continuous embeddings, we apply the same principle to the pre-binarization logits that parameterize the discrete communication channel. Concretely, we promote effective utilization of the discrete bottleneck by optimizing a coding-rate objective on the L2-normalized logits $a$, encouraging them to be well spread on the unit hypersphere:

$$\mathcal{L}_{rate} = -R_\epsilon(a) = -\frac{1}{2} \log \det \left( I + \frac{B}{\epsilon^2} A \right), \tag{10}$$
$$\text{s.t.} \quad A = \text{Cov}[a/\|a\|_2].$$

### 4.4. Final Objective

The full final training objective is:

$$\mathcal{L} = \mathcal{L}_{BCE} + \beta \times \mathcal{L}_{rate}, \tag{11}$$

where $\beta$ is a hyperparameter. Both branches are initialized similarly, the student branch is updated through gradi-

ent backpropagation of the loss $\mathcal{L}$, and the teacher branch through exponentially moving average (EMA). Together, $\mathcal{L}_{BCE}$ and $\mathcal{L}_{rate}$ provide a practical optimization of the two complementary terms of the mutual information objective in Eq. 4: invariance and capacity utilization.

### 4.5. Periodically Randomized Heads

While discrete agreement and coding-rate regularization encourage informative and factorized representations, keeping the projection head fixed throughout training may bias the backbone toward a specific parameterization of the discrete code space. To reduce this dependency and further promote robustness to the choice of coding scheme, we periodically reinitialize the projection head $g_\phi$ every $n$ epochs by resampling its parameters from a fixed initialization distribution $\mathcal{D}$.

Each reset induces a new random mapping from the backbone to the binary code space, encouraging the backbone to learn features that remain predictive across multiple discrete encodings rather than adapting to a single fixed mapping. From a communication viewpoint, each projection head can be interpreted as a distinct parameterization of the discrete channel, and exposing the backbone to multiple such parameterizations promotes representations that make more effective use of the fixed-capacity discrete bottleneck.

## 5. Experiments

We follow the SimDINO setup, and by extension DINO, as closely as possible to ensure a fair comparison. Experiments are conducted using a Vision Transformer backbone (ViT-B/16) trained from random initialization, together with the same three-layer MLP projection head as in DINO. The projection head outputs $B = 256$ units, which corresponds to the default dimensionality used in DINO and SimDINO; this value is not tuned for our method and is kept identical across all models. In our framework, these outputs parameterize a 256-bit binary message in the discrete communication channel. Standard self-supervised data augmentations are used to generate multiple views, with local views fed to the student branch.

All models are trained using the official SimDINO implementation as a common optimization framework, ensuring identical architectures, optimization settings, and data augmentations across methods. Models are trained for 100 epochs on ImageNet-1K with a batch size of 256. For our method, the projection head parameters $\phi$ are periodically re-initialized during training. We reset the projection head every $n = 10$ epochs, a moderate frequency that we found to provide a good trade-off between stability and diversity; a detailed ablation on $n$ is reported in Appendix A.

We did not conduct an in-depth comparison with DINOv2

(or v3), as it introduces a patch-level training objective that is not directly compatible with our binary agreement framework. In particular, applying a multi-label loss at the patch level is not necessarily desirable, as local patches do not naturally admit a multi-label semantic interpretation. While it would be possible to restrict the binary agreement objective to the global views only, this would result in a hybrid setup that complicates both evaluation and interpretation, and would blur the core contribution of this work. Although it is not the main focus of this paper, we provide some comparisons in Appendix E for reference.

In the remainder of the paper, we refer to our discrete communication framework as BITS (Binary Information Transmission for Self-Supervision).

### 5.1. ImageNet-1k Classification and Retrieval

We evaluate the learned continuous backbone representations on ImageNet-1K using two complementary tasks: classification and retrieval. Classification assesses global invariance and semantic alignment, while retrieval is particularly sensitive to the structure of the embedding space, as it relies on fine-grained similarity rather than class-level separation. This makes retrieval a natural benchmark for evaluating the benefits of factorized and multi-label representations. We report k-NN classification, linear probing accuracy, and mAP@ALL on the retrieval split of FFPQ (Liang et al., 2023).

We compare **DINO** (softmax-based agreement with a fixed continuous head), **SimDINO** (cosine-based agreement with a fixed continuous head), **BITS-fixed** (binary agreement with a fixed head), and **BITS-reset** (binary agreement with periodic head reinitialization every $n = 10$ epochs). Results are reported in Tab. 1.

*Table 1.* Classification and retrieval performance comparison on ImageNet-1k.

| Method | mAP | k-NN | linear probing |
|---|---|---|---|
| DINO | 35.68 | 72.39 | 76.3 |
| SimDINO | 38.62 | 69.52 | 75.3 |
| BITS-fixed | 43.44 | 73.32 | 76.7 |
| BITS-reset | **50.64** | **73.5** | **77.8** |

Overall, both BITS variants outperform DINO and SimDINO across all metrics, with **BITS-reset** achieving the strongest performance. The largest relative gains are observed for retrieval, highlighting the benefit of multi-label agreement for tasks that rely on fine-grained similarity. These results indicate that discretizing the agreement mechanism, rather than the representation itself, induces more structured embeddings and yields consistent improvements

over SimDINO, despite both approaches relying on coding-rate regularization. The additional gains of BITS-reset over BITS-fixed further show that periodic head reinitialization helps fully exploit the backbone capacity under a discrete agreement objective.

### 5.2. Representation Geometry and Expressivity

To better understand the gains observed for retrieval, we next analyze the geometry and expressivity of the learned representations. We first study how variance is distributed across backbone dimensions following (Kim et al., 2025; Li et al., 2022a). We extract continuous backbone features from the ImageNet-1K validation set and compute the empirical covariance matrix, whose eigenvalues $\lambda_1 \geq \cdots \geq \lambda_d$ are used to estimate the cumulative explained variance, computed for the first $i$ dimensions as: $cv_i = \sum_{j=1}^{i} \lambda_j \big/ \sum_{j=1}^{d} \lambda_j$.

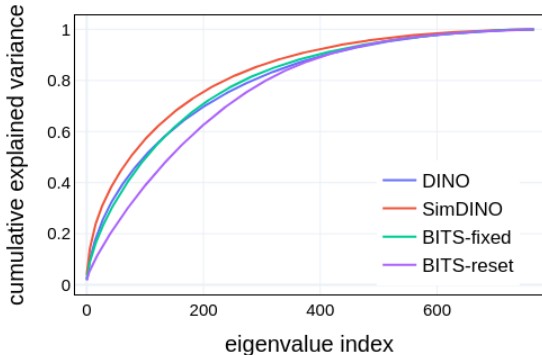

*Figure 2.* Cumulative explained variance of the dimensions of the representations of ImageNet-1k validation set.

Fig. 2 reports the cumulative explained variance as a function of the number of backbone dimensions for all compared methods. Despite both relying on coding–rate regularization, SimDINO's cosine-based agreement concentrates variance more strongly along a small number of dominant dimensions than BITS-fixed. In contrast, the discrete multi-label agreement used in BITS-fixed promotes a more balanced allocation of variance across dimensions.

DINO exhibits a variance distribution comparable to BITS-fixed, indicating that enforcing a mono-label agreement already encourages a degree of factorization. However, this factorization arises from prototype selection and remains inherently limited to a single active semantic mode per sample. Finally, BITS-reset further improves variance uniformity, suggesting that periodically reinitializing the projection head encourages the backbone to distribute information more evenly by preventing adaptation to a single fixed coding scheme.

While variance distribution characterizes how information is allocated across dimensions, it does not capture how much effective representational capacity is uti-

lized. To quantify expressivity, we compute the effective representation dimension $d_{eff} = \left(\sum_j \lambda_j\right)^2 / \sum_j \lambda_j^2$ and the effective representation rank $r_{eff} = \exp\left(-\sum_j \tilde{\lambda}_j \log \tilde{\lambda}_j\right)$ where $\tilde{\lambda}_j = \lambda_j / \sum_k \lambda_k$ in Tab. 2.

While DINO and BITS-fixed achieve comparable levels of factorization, BITS-fixed consistently attains higher effective dimensionality, reflecting the increased expressivity enabled by multi-label representations. While

Table 2. Dimensional collapse comparison between models.

| Method | $d_{eff}$ | $r_{eff}$ |
|--------|-----------|-----------|
| DINO | 233 | 400 |
| SimDINO | 156 | 324 |
| BITS-fixed | 250 | 404 |
| BITS-reset | **358** | **480** |

mono-label agreement promotes separation across samples, multi-label discrete agreement allows multiple semantic factors to be simultaneously active, increasing the number of distinguishable configurations.

BITS-reset further increases both $d_{eff}$ and $r_{eff}$, indicating a more effective utilization of the fixed-capacity discrete channel. This suggests that periodic head reinitialization not only promotes factorization, but also encourages the backbone to exploit the full combinatorial expressivity of the binary representation space.

### 5.3. Ablations Studies

In this subsection, we perform multiple ablations to test different components of our framework.

**Binary cross-entropy vs. cosine agreement on logits.** We compare two agreement formulations within the same discrete binary channel: a per-bit binary cross-entropy loss and a cosine similarity applied to the student logits and thresholded teacher outputs (with targets in $\{-1, +1\}$). Both variants use identical architectures and channel dimensionality. Tab. 3 shows that enforcing agreement independently for each bit leads to substantially better retrieval performance than using a global cosine similarity, despite identical discretization and capacity. This indicates that discretization alone is not sufficient: effectively exploiting a multi-label binary channel requires an agreement objective that operates at the level of individual dimensions. In contrast, cosine similarity treats the binary message as a single vector and allows compensations across bits, resulting in weaker utilization of the discrete representation.

Table 3. Effect of agreement formulation within a discrete binary channel.

| Agreement | BCE | Cosine |
|-----------|-----|--------|
| mAP | **43.44** | 36.34 |

**Hard vs. soft binary targets.** We compare soft continuous targets with hard binary targets obtained via thresholding in the discrete communication framework. Without temperature tuning, soft targets obtained through sigmoid activations lead to unstable optimization and fail to reliably converge. When the temperature is carefully tuned, the best-performing soft-target regime (e.g., $\tau = 0.1$) yields performance close to hard binary targets by producing near-binary labels. However, this regime is narrow, nearly binary, and highly sensitive to the chosen value (see Appendix A). In contrast, hard binary targets directly provide strictly binary labels in a deterministic and robust manner, yielding a stable supervision signal without additional tuning.

**Coding-rate control.** We analyze the effect of the coding-rate coefficient $\beta$, which controls the utilization of the discrete binary channel. We report retrieval performance after 10 training epochs for different fixed values of $\beta$ in Tab. 4. Extremely small or large values hinder effective channel usage, while $\beta = 0.1$ yields stable optimization and is used in all subsequent experiments.

Table 4. Effect of the coding rate regularization coefficient $\beta$ after 10 epochs of training.

| $\beta$ | 0.05 | 0.1 | 0.2 | 0.5 |
|---------|------|-----|-----|-----|
| mAP | 0.22 | **10.46** | 7.21 | 1.1 |

**Periodically randomized heads on cosine vs. on BCE.** Applying periodic head resets to SimDINO (cosine-based SSL) leads to unstable training and divergence, whereas the same procedure remains stable under binary cross-entropy agreement. This highlights a key difference between global cosine alignment and per-bit binary supervision: cosine-based objectives are sensitive to abrupt changes in teacher representations, while the bounded per-dimension gradients induced by BCE make discrete agreement more robust to head reinitialization.

Overall, these ablations indicate that discrete agreement is driven by per-bit binary supervision with hard thresholding, with coding-rate regularization controlling channel utilization; head reinitialization is an optional optimization enabled by discretization.

### 5.4. Downstream Transfer under Domain Shift

We assess the transferability of BITS pre-trained backbone to downstream tasks under varying degrees of domain shift (using frozen representations unless stated otherwise).

**From In-Domain to Light Domain Shift Retrieval.** We first evaluate retrieval performance under in-domain and light domain shift using ImageNetV2, ImageNet100, PascalVOC2012 (Everingham et al.), and COCO2014 (Lin et al., 2014). These datasets progressively depart from the

ImageNet-1k training distribution while preserving similar object-level semantics. As reported in Tab. 5, both BITS variants consistently outperform the continuous baselines across all settings, indicating that discrete binary agreement preserves transferable semantic structure beyond the source domain.

*Table 5.* Retrieval performance using mAP on in-domain and light out-of-distribution datasets.

| Method | INv2 | IN100 | Pascal | Coco |
|---|---|---|---|---|
| DINO | 38.77 | 76.11 | 75.84 | 83.59 |
| SimDINO | 41.48 | 72.86 | 75.49 | 82.9 |
| BITS-fixed | 46.15 | 79.44 | 78.34 | **85.4** |
| BITS-reset | **52.9** | **82.29** | **79** | 85.31 |

**Unsupervised Object Detection and Segmentation.** We further assess transferability on object-centric tasks using MaskCut (Wang et al., 2023) for unsupervised object detection and instance segmentation on COCO2017 (Lin et al., 2014). As shown in Tab. 6, both BITS variants achieve stronger performance than DINO and SimDINO, confirming that the learned representations capture complementary object-level semantics, despite not being explicitly trained for dense prediction.

*Table 6.* Unsupervised object detection and segmentation via Mask-Cut evaluated on COCO val2017.

| Method | Detection | | | Segmentation | | |
|---|---|---|---|---|---|---|
| | $AP_{50}$ | $AP_{75}$ | AP | $AP_{50}$ | $AP_{75}$ | AP |
| DINO | 3.9 | 1.5 | 1.8 | 3.1 | 1 | 1.4 |
| SimDINO | 5 | 1.6 | 2.2 | 4 | 1.4 | 1.6 |
| BITS-fixed | 6 | **2.5** | **2.8** | 5 | **1.8** | **2.2** |
| BITS-reset | **6.4** | 2.2 | **2.8** | **5.3** | 1.6 | **2.2** |

**Video Object Segmentation.** We evaluate local patch representations on DAVIS2017 (Pont-Tuset et al., 2017) using nearest-neighbor propagation between consecutive frames, following the DINO evaluation protocol. Results in Tab. 7 show that discrete agreement preserves spatially coherent patch features, despite not being explicitly optimized for dense prediction.

*Table 7.* Semi-supervised object segmentation on Davis 2017 Video dataset.

| Method | $(\mathcal{J}\&\mathcal{F})_m$ | $\mathcal{J}_m$ | $\mathcal{F}_m$ |
|---|---|---|---|
| DINO | 61.83 | 60.44 | 63.21 |
| SimDINO | 61.98 | 60.06 | 63.9 |
| BITS-fixed | **62.78** | **60.99** | **64.56** |
| BITS-reset | 62.51 | 60.81 | 64.21 |

**Severe Domain Shift: Fine-grained Recognition.** We

evaluate transfer under severe out-of-distribution shift using Birds525 (Piosenka, 2023), Food101 (Bossard et al., 2014), iNat2019 (Horn et al., 2019) and PlantNet300k (Garcin et al., 2021), four fine-grained and highly domain-specific datasets relying on localized visual cues. Given the severity of the domain shift and the fine-grained nature of the tasks, we focus on linear probing accuracy, which provides a standard protocol for evaluating semantic transfer in this setting; additional metrics are reported in Appendix C.

*Table 8.* Linear probing accuracy on fine-grained out-of-distribution datasets.

| Method | Birds525 | Food101 | iNat2019 | PlantNet |
|---|---|---|---|---|
| *Frozen ImageNet-1k pre-trained weights* | | | | |
| DINO | 87.05 | 75.59 | 36.63 | 67.66 |
| SimDINO | 92 | 75.79 | 36.2 | 67.18 |
| BITS-fixed | **95.66** | **81.52** | **53.2** | **72.16** |
| BITS-reset | 84.15 | 74.78 | 29.74 | 63.45 |
| *Self-supervised fine-tuned on target domain* | | | | |
| DINO | 77.52 | 70.9 | 24.82 | 71.79 |
| SimDINO | 91.09 | 66.19 | 29.08 | 70.04 |
| BITS-fixed | **96.72** | **82.88** | **54.39** | **80.04** |
| BITS-reset | 95.35 | 82.69 | 49.01 | 79.1 |

As shown in Tab. 8, clear differences emerge between agreement mechanisms under severe domain shift. Among frozen models, BITS-fixed achieves the strongest linear probing performance, indicating that discrete binary agreement with a fixed coding scheme yields robust and linearly separable semantic factors that generalize across fine-grained domains. In contrast, BITS-reset underperforms in the frozen setting, revealing a robustness–factorization trade-off: periodic head resets encourages highly factorized but more source-specific representations. When self-supervised fine-tuning on the target domain is allowed, this gap is largely closed, with BITS-reset, reinitialized only once at the start of the 10-epoch adaptation, reaching performance comparable to BITS-fixed. By contrast, continued self-supervised training of continuous baselines does not consistently improve performance and can even degrade accuracy. We hypothesize that our structured information allocation is what explains the behavior under domain-shift fine-tuning. When representations are more factorized, changes affect only a subset of dimensions, making adaptation easier. However, as shown in (Locatello et al., 2019), classical self-supervised methods do not encourage decomposition of semantic factors, and as a result, their entangled representations mix factors, so even small shifts can impact the representation globally.

### 5.5. Analysis of the Learned Binary Language

Our framework introduces discrete communication as a self-supervised pretext task, inducing explicit binary codes at the output of the projection head. Although these codes are not used directly at inference time, analyzing their struc-

ture provides insight into how discrete agreement organizes semantic information.

Fig. 3 visualizes the learned binary codes on INet10 (Howard, 2019) using t-SNE with Hamming distance, embedding the codes in three dimensions and visualizing different two-dimensional projections. Across projections, the embeddings display a recurring ring-like structure, with samples distributed at comparable Hamming distances, while avoiding collapse or domination by a small subset of bits. This geometry is consistent with a balanced use of the fixed-capacity binary channel induced by the discrete communication objective.

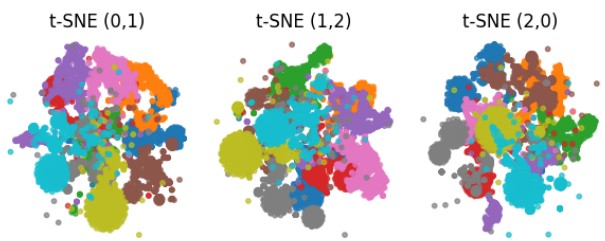

*Figure 3.* Visualization of t-SNE embeddings of the hash codes.

To quantify the information content of the codes, we measure their entropy. The average marginal entropy per bit is approximately $0.9$, indicating that most bits are active and well balanced. More strikingly, the entropy computed over contiguous 8-bit blocks reaches approximately $7.1$ bits out of a maximum of $8$. This result goes beyond marginal balance and indicates that the discrete communication objective promotes high joint entropy over groups of bits, rather than concentrating information independently or redundantly across dimensions. We further evaluate the usefulness of the learned binary codes for retrieval. Using the full 256-bit representation yields an mAP of $47.59$, compared to $50.64$ when using the continuous backbone embeddings. Subsampling the code to 128, 64, and 32 bits results in a gradual performance decrease ($45.97$, $43.73$, and $40.35$, respectively). The relatively smooth degradation indicates that retrieval performance does not rely on a small subset of bits, but instead draws on information distributed across the binary channel, consistent with the high joint entropy observed in the code analysis.

To qualitatively analyze the semantic structure of the learned binary codes, we visualize images conditioned on the value of bit number 0 in Fig. 4 (additional examples are reported in Appendix D). For this bit, we observe a consistent separation between images depicting objects in isolation and images showing objects embedded in a scene with human presence or interaction. Importantly, this behavior is shared across semantically distinct classes, such as *tench* and *french horn*, indicating that the bit does not encode class-specific information but rather captures a higher-level contextual

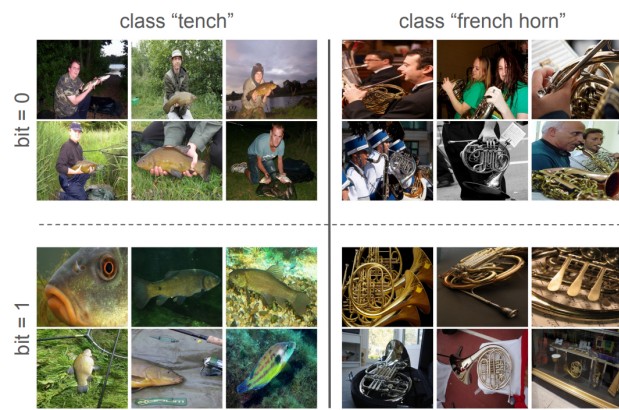

*Figure 4.* Visualization of images from two ImageNet classes conditioned on the value of bit 0 in each row. Top row (bit=0) contains humans. Bottom row (bit=1) consistently shows no humans.

attribute. This suggests that individual bits can represent reusable semantic cues related to scene context or object usage, which are shared across classes.

### 5.6. Limitations

Our experiments are conducted exclusively at the ViT-B/16 scale, as our primary goal is to isolate the effect of the proposed discrete communication objective in a controlled and widely used setting, enabling a direct comparison of training objectives without confounding factors related to scale or architecture. Evaluating the proposed objective at larger scales, such as ViT-L or ViT-g, is an interesting direction for future work.

## 6. Conclusion

We introduced a discrete communication perspective on self-supervised learning, where semantic consistency is enforced through agreement over a fixed-capacity binary channel rather than continuous similarity. By aligning teacher and student representations through binary messages, our approach induces structured, multi-label representations that are more factorized than those learned with standard continuous agreement objectives, while remaining competitive or superior across a wide range of downstream tasks. Beyond the specific instantiation studied in this work, our results suggest that discrete communication offers a general and flexible principle for representation learning. Future work may explore richer discrete languages beyond binary codes, such as learned vocabularies or symbolic tokens, as well as extensions to other self-supervised settings, including multimodal alignment and temporal modeling.

## Impact Statement

This paper presents work whose goal is to advance the field of machine learning. There are many potential societal consequences of our work, none of which we feel must be specifically highlighted here.

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

# A. Supplementary Ablations.

Appendix A presents complementary ablation studies that further justify the design choices of our method and analyze their impact beyond the settings reported in the main text. Unless stated otherwise, ablations follow the same training protocol as the main experiments (100 epochs), ensuring that observed differences reflect relative trends rather than changes in training budget.

**Head reset frequency.** This ablation further characterizes the optional head reinitialization mechanism introduced in the main paper. In particular, We study the effect of the projection head reset frequency $n$ on retrieval performance. As shown in Tab. 9, periodically resetting the head significantly improves the model compared to keeping a fixed head ($n = \infty$), which shows that periodically changing the parameterization of the discrete channel allows the backbone to further adapt its representations to the source domain, beyond what is achieved with a fixed coding scheme. Resetting too frequently degrades performance, likely because the model does not have sufficient time to adapt to each coding scheme.

*Table 9.* Effect of head reset frequency $n$.

| $n$ | 1 | 5 | 10 | 20 | $\infty$ |
|---|---|---|---|---|---|
| mAP | 45.21 | 49.68 | **50.64** | 50.47 | 43.44 |

**Hard vs. soft binary targets.** These experiments further justify the target construction choice adopted in the main experiments, namely hard thresholding at 0.5. This choice naturally follows from the Bernoulli parameterization and corresponds to maximum a posteriori decoding under a symmetric prior. We compare this strategy with soft continuous targets obtained via sigmoid activation. As shown in Tab. 10, soft targets lead to unstable optimization at high temperatures, with the binary cross-entropy loss failing to converge. We attribute this behavior to the ambiguity and instability of the targets across training iterations. When using lower temperatures, the model operates closer to the thresholding regime and can converge. We hypothesize that this is because such temperatures already lead to nearly binary targets. In practice, we analyze the actual target distribution and find that, for instance, at $\tau = 0.1$, $48.33\%$ of the values are $> 0.99$ and $48.25\%$ are $< 0.01$, and at $\tau = 0.01$, this increases to $49.66\%$ and $49.99\%$ respectively. However, performance remains sensitive to temperature tuning and consistently slightly inferior. In contrast, deterministic hard thresholding yields stable optimization and the best performance, providing a stronger and more reliable supervision signal under the discrete agreement objective.

*Table 10.* Effect of hard vs. soft targets with different temperatures $\tau$.

| target | soft | | | | | hard |
|---|---|---|---|---|---|---|
| | $\tau = 1$ | $\tau = 0.2$ | $\tau = 0.1$ | $\tau = 0.05$ | $\tau = 0.01$ | |
| mAP | collapse | 43.06 | 43.28 | 43.06 | 43.19 | **43.44** |

**Number of bits B.** We examine how the dimensionality of the the projection head affects our model. As reported in Tab. 11, using too few bits limits the representation capacity of our model. In contrast, a larger number of bits decreases performance, which we hypothesize is due to over-parameterization and optimization difficulty, especially of the regularization objective. $B = 256$ provides a good compromise between capacity and trainability. Notably, $B = 256$ corresponds to the default projection dimensionality used in DINO-style frameworks, suggesting that this choice reflects a broadly effective operating point rather than dataset-specific or agreement-specific tuning.

*Table 11.* Effect of the number of bits $B$.

| $B$ | 128 | 256 | 512 |
|---|---|---|---|
| mAP | 38.46 | **43.05** | 39.32 |

**Gradient clipping.** Tab. 12 shows the effect of varying the clipping threshold $cg$ (maximum gradient norm) on the model performance. Performance remains relatively stable across a broad range of clipping value with $cg = 1$ providing the best performance.

*Table 12.* Effect of gradient clipping value $cg$.

| $cg$ | 0.3 | 0.5 | 1 | 2 | 3 |
|------|------|------|------|------|------|
| mAP | 42.52 | 42.69 | **43.05** | 42.78 | 42.68 |

**End-of-training learning rate.** We find that the choice of the minimum learning rate achieved at the end of training affects the model performance, as shown in Tab. 13. As training progresses, BCE saturates rapidly and gradients vanish near convergence, making updates extremely small. This behavior reflects a general property of binary cross-entropy objectives near convergence, rather than a limitation specific to our discrete agreement formulation. Setting a slightly higher $min\_lr$ allows the model to continue making meaningful weight updates.

*Table 13.* Effect of end-of-training learning rate $min\_lr$.

| $min\_lr$ | $10^{-6}$ | $5 \times 10^{-5}$ |
|------|------|------|
| mAP | 43.05 | **43.44** |

## B. Self-supervised Training Curves.

Fig. 5 reports the evolution of validation retrieval performance during SSL training for continuous agreement baselines and our discrete agreement variants. The figure shows that our discrete binary agreement models exhibit faster convergence and consistently higher retrieval performance throughout training. While DINO exhibits a strong start, its convergence gradually slows down with epochs. In contrast, SimDINO starts lower but rapidly accelerates, gaining important convergence speed. On the other hand, our models both benefit from a substantial head start and maintain a consistently strong convergence rate. In particular, BITS-reset shows a consistent improvement, which highlights the benefit of periodic head reset for in-domain specialization. This behavior further showcases that discrete binary agreement provides a stronger supervisory signal early in training and leads to more structured representations as training progresses.

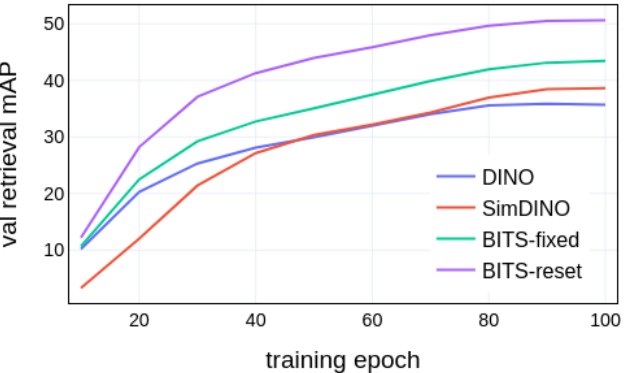

*Figure 5.* Pre-training on ImageNet-1k: mAP score evolution across epochs.

## C. More on Adaptation to Severe Domain Shift.

This appendix complements the *Severe Domain Shift* analysis of the main paper by reporting additional evaluation metrics, including mAP and k-NN classification alongside linear probing, and by extending the study to an additional fine-grained dataset.

We report in Tab. 14 results on Birds525 (Piosenka, 2023), a dataset that still shares partial semantic overlap with ImageNet, Food101 (Bossard et al., 2014), iNat2019 (Horn et al., 2019), and PlantNet300k (Garcin et al., 2021), that are highly specialized.

Table 16 reports retrieval (mAP), k-NN classification, and linear probing results on four fine-grained datasets under severe domain shift. Overall, frozen mAP and k-NN performance remain relatively low and unstable across methods, with substantial variability across datasets. This behavior is expected in fine-grained recognition settings, where classes are separated by subtle visual cues and exhibit strong inter-class ambiguity. In such regimes, global semantic structure and

*Table 14.* Retrieval and classification performance on different out-of-distribution datasets.

| Method | Birds525 | | | Food101 | | | iNat2019 | | | PlantNet300k | | |
|---|---|---|---|---|---|---|---|---|---|---|---|---|
| | mAP | k-NN | LP | mAP | k-NN | LP | mAP | k-NN | LP | mAP | k-NN | LP |
| *Pre-trained out-of-domain ImageNet-1k weights* | | | | | | | | | | | | |
| DINO | 38.86 | 86.32 | 87.05 | 14.6 | 65.16 | 75.59 | 12.29 | 35.25 | 36.63 | 12.11 | **58.79** | 67.66 |
| SimDINO | **57.07** | **93.52** | 92 | 17.5 | **66.2** | 75.79 | **12.87** | **37.1** | 36.2 | **12.13** | 57.88 | 67.18 |
| BITS-fixed | 46.45 | 87.81 | **95.66** | **19.82** | 65.49 | **81.52** | 12.27 | 36.47 | **53.2** | 11.44 | 56.94 | **72.16** |
| BITS-reset | 31.46 | 73.71 | 84.15 | 16.05 | 53.53 | 74.78 | 8.76 | 23.83 | 29.74 | 5.93 | 43.05 | 63.45 |
| *Self-supervised finetuned in-domain weights* | | | | | | | | | | | | |
| DINO | 34.94 | 81.25 | 77.52 | 22.1 | 65.89 | 70.9 | 14.24 | 35.38 | 24.82 | 32.03 | 70.44 | 71.79 |
| SimDINO | 57.18 | 92.61 | 91.09 | 13.73 | 57.67 | 66.19 | 14.57 | 35.97 | 29.08 | 19.48 | 63.08 | 70.04 |
| BITS-fixed | 59.86 | 91.62 | **96.72** | 33.17 | 75.67 | **82.88** | 19 | 40.59 | **54.39** | 34.8 | 72.12 | **80.04** |
| BITS-reset | **70.52** | **93.9** | 95.35 | **40.02** | **76.61** | 82.69 | **21.18** | **40.89** | 49.01 | **38.2** | **72.47** | 79.1 |

invariances learned during pre-training on other domains are often insufficient to support reliable retrieval or nearest-neighbor classification without adaptation. While some methods, such as SimDINO, achieve higher frozen mAP or k-NN scores on some datasets (e.g., Birds525), these trends do not translate into improved linear probing performance. Linear probing provides a relevant evaluation protocol under severe domain shift, as it explicitly allows the model to reweight and recombine learned features to resolve fine-grained distinctions.

Under continued self-supervised fine-tuning on the target domain, BITS variants exhibit clear and consistent performance improvements across evaluation metrics. As discussed in the main paper, both BITS-fixed and BITS-reset achieve stronger linear probing accuracy than continuous baselines, confirming that discrete agreement provides a more effective inductive bias for adapting representations under severe domain shift. In addition to these gains in linear probing, continued self-supervised adaptation also improves frozen retrieval and k-NN classification performance. In this setting, BITS-reset achieves the strongest results for mAP and k-NN classification, while BITS-fixed remains superior in linear probing. This complementary behavior reflects a trade-off between specialization and linear accessibility: periodic head reinitialization favors adaptation of the representation geometry for retrieval-based metrics, whereas a fixed discrete coding scheme preserves more linearly separable features for fine-grained classification.

## C.1. Probing Curves.

Figures 6 and 7 report respectively the linear probing and MLP probing performance on Birds525 and PlantNet300k as a function of the number of training iterations. The curves show that BITS-fixed exhibits a clear and consistent advantage from the very first iterations, indicating that the learned representations are immediately more linearly accessible under severe domain shift. This early performance gap persists throughout training and is observed for both linear and shallow nonlinear probes, further supporting the robustness of the semantic structure induced by discrete agreement.

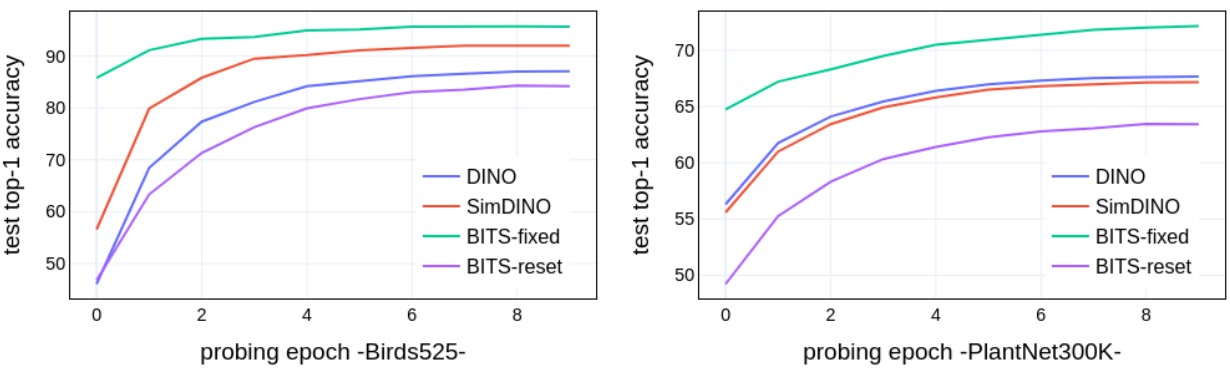

*Figure 6.* Linear probing results evolution on Birds525 (left) and PlantNet300k (right).

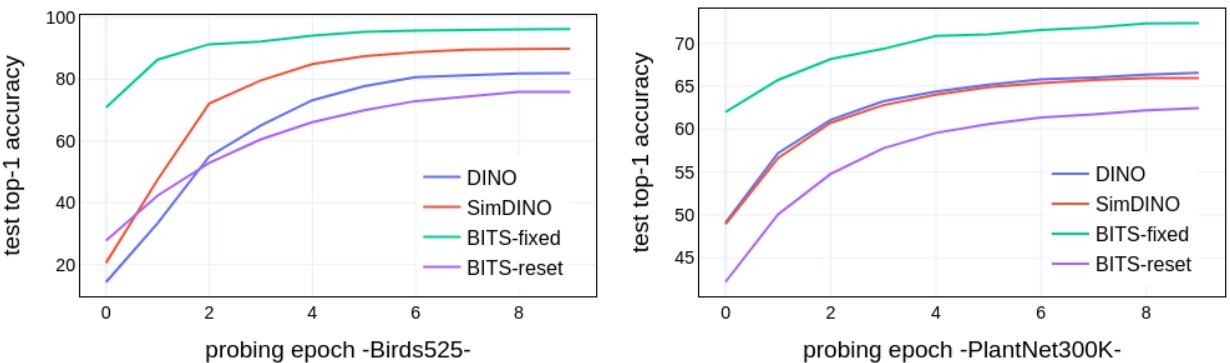

*Figure 7.* MLP probing results evolution on Birds525 (left) and PlantNet300k (right).

## C.2. Continued SSL Finetuning Curves.

Figure 8 reports the evolution of downstream retrieval performance during continued self-supervised fine-tuning on Birds525 and PlantNet300k. The curves show that BITS-reset achieves a markedly faster performance increase during adaptation, highlighting the benefit of head reinitialization for rapid specialization under severe domain shift.

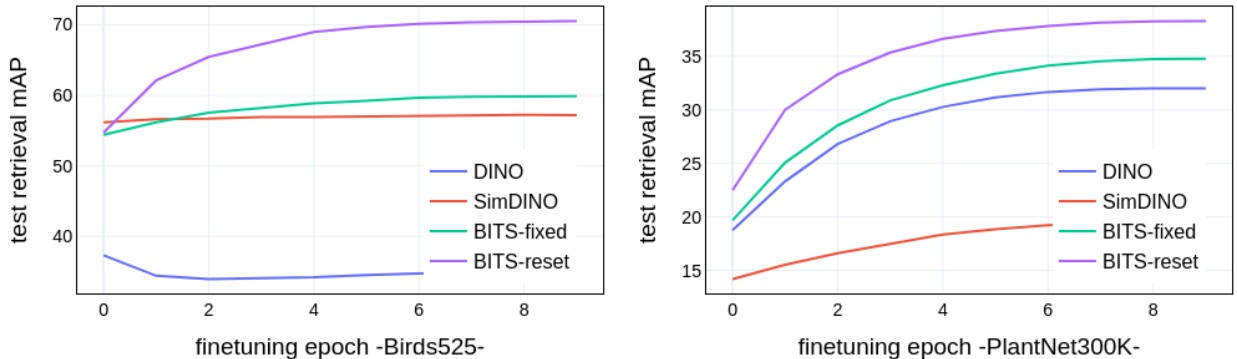

*Figure 8.* Full self-supervised finetuning results evolution on Birds525 (left) and PlantNet300k (right).

## D. More Qualitative Results.

We present additional qualitative visualization in Figs. 9, 10, 11. For three different bits, we visualize ImageNet-1k samples from different classes, conditioned on the bit value. Across all cases, we observe that individual bits activate for specific semantic attributes that are shared across classes, rather than encoding class-specific information. Samples activating a given bit exhibit coherent visual properties—such as contextual, structural, or object-level cues—that remain stable across different categories. These visualizations illustrate that the learned binary codes capture reusable semantic attributes that generalize beyond individual classes, supporting a compositional representation of visual concepts. This trans-class consistency highlights the structured and interpretable nature of the discrete language induced by the proposed discrete communication objective.

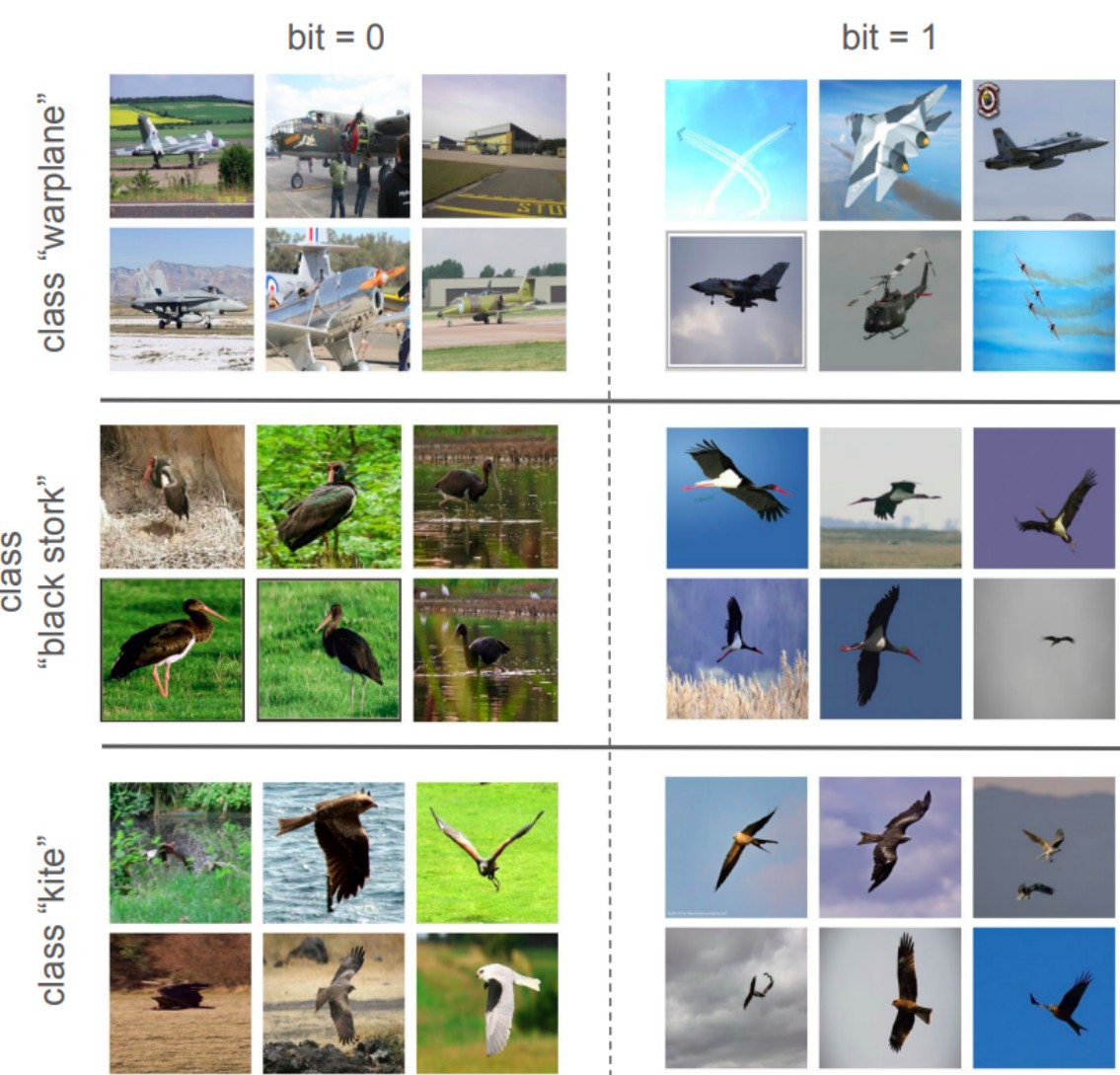

*Figure 9.* Visualization of images from three ImageNet classes (*warplane*, *black stork*, *kite*) conditioned on the value of bit 17.

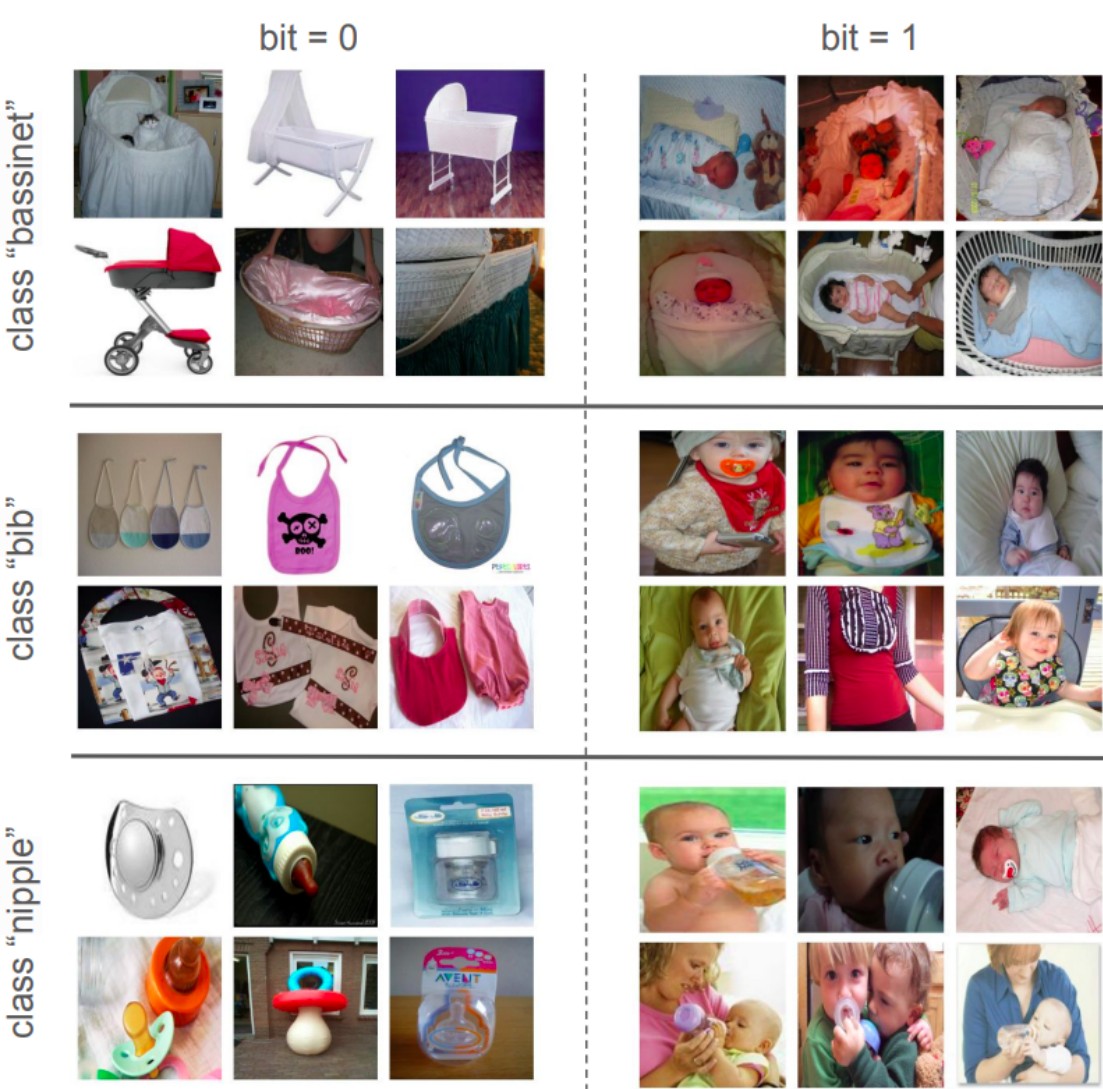

*Figure 10.* Visualization of images from three ImageNet classes (*bassinet*, *bib*, *nipple*) conditioned on the value of bit 2.

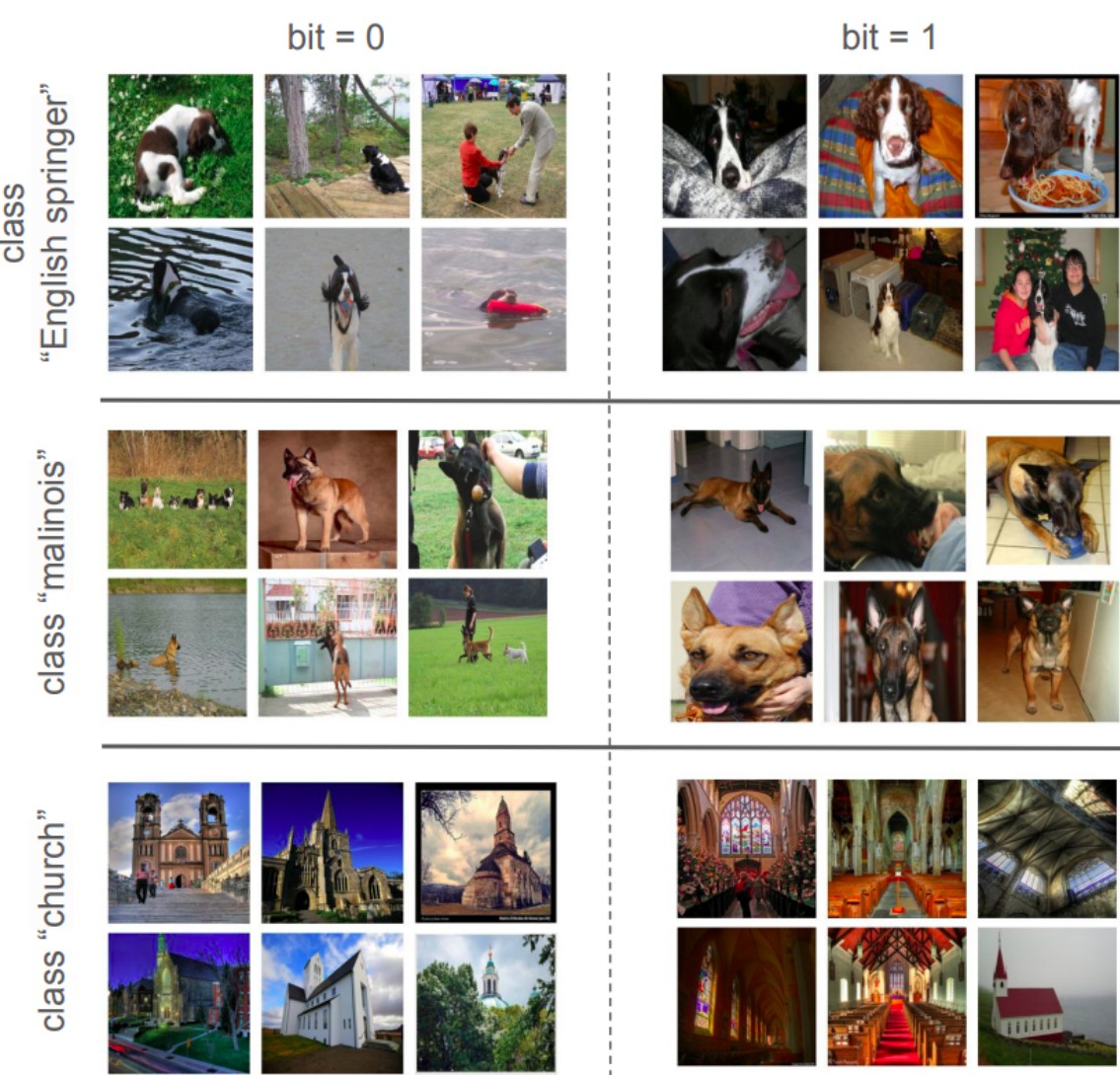

*Figure 11.* Visualization of images from three ImageNet classes (*English springer*, *malinois*, *church*) conditioned on the value of bit 121.

# E. Comparison with More Complex Architectures.

The primary goal of this work is to isolate the effect of the discrete communication objective in a controlled setting, using a fixed architecture. Comparisons with methods that differ along multiple axes simultaneously, such as architecture size, training duration, or additional objectives, are therefore not the main focus of this paper. Nevertheless, for completeness, we report in Tab. 15 additional comparisons with DINOv2 and a hybrid variant of our method. BITS-fixed already outperforms DINOv2 in retrieval while being slightly lower in k-NN and linear probing. BITS-reset further improves retrieval and linear probing, while remaining slightly lower in k-NN. These results suggest that the discrete communication objective is competitive even when compared to methods trained with additional objectives and longer schedules, though a fully controlled comparison would require matching all training factors. We also explore combining our discrete communication framework with an iBOT-style patch-level objective, using a binary cross-entropy formulation with multi-bit discrete targets (8 bits per patch). This hybrid variant consistently improves over DINOv2 across all metrics, indicating that discrete communication is not only competitive as a standalone objective but also complementary to local patch-level losses. Exploring this direction more systematically is left for future work.

*Table 15.* Classification and retrieval performance comparison on ImageNet-1k.

| Method | mAP | k-NN | linear probing |
|---|---|---|---|
| DINOv2 | 38.33 | 75.93 | 76.8 |
| BITS-fixed | 43.44 | 73.32 | 76.7 |
| BITS-reset | 50.64 | 73.5 | 77.8 |
| iBOT-style hybrid variant | 47.03 | 76.66 | 78 |

