# OpenReview forum: "Self-Supervised Learning as Discrete Communication"
_ICML.cc/2026/Conference — ICML 2026 regular_

### Official Review · Reviewer_JtoP · 2026-02-14

**Soundness:** 3
**Presentation:** 3
**Significance:** 2
**Originality:** 3
**Overall Recommendation:** 4
**Confidence:** 4

**Summary:**

This paper introduces a framework that reformulates self-supervised learning as a discrete communication process. Instead of enforcing consistency through continuous similarity matching, BITS transmits semantic information via a fixed-capacity binary channel, where the student network predicts multi-label binary messages generated by the teacher network.
This approach aims to combine the performance advantages of continuous alignment methods with the structural benefits of discrete representations. By enforcing a binary bottleneck and periodically reinitializing the projection head, the method encourages the learning of factorized and compact representations. In multiple benchmarks, including image classification and retrieval, BITS achieves consistent performance improvements relative to continuous alignment baselines.

**Compliance With Llm Reviewing Policy:**

Affirmed.

**Final Justification:**

We thank the authors for their clear and well-structured rebuttal. The three main concerns have been satisfactorily addressed.

**Key Questions For Authors:**

1. You explicitly choose to apply the discrete bottleneck only at the very end of the MLP projection head. Have you explored applying discretization at earlier stages (e.g., directly on the backbone features or within intermediate MLP layers)? Does discretizing only the final output limit the model's ability to learn hierarchical discrete structures, or is it sufficient to propagate this structure back to the continuous backbone?

2. The paper frames the method as maximizing information transmission through a fixed-capacity binary channel." However, the argument relies heavily on empirical results rather than information-theoretic analysis. Can you provide any theoretical analysis?

**Limitations:**

1. Explicitly acknowledging the lack of formal information-theoretic guarantees for the channel capacity claims, which are currently justified empirically.

2. It is unclear if discrete binary representations can effectively align with continuous text embeddings for downstream tasks such as Visual Question Answering (VQA) and Image Captioning. These tasks rely heavily on granular, continuous semantic alignment, which might be compromised by the hard discretization.

**Strengths And Weaknesses:**

1. The paper demonstrates consistent improvements over strong baselines (SimDINO, DINO) across a wide range of tasks, including image classification, retrieval, and dense prediction. The gains in retrieval tasks are particularly noteworthy, suggesting that the discrete bottleneck effectively structures the embedding space.

2. The inclusion of experiments on domain shift and self-supervised adaptation significantly strengthens the paper's claims. These results indicate that the learned features are not merely overfitting to the source domain but are capturing more fundamental semantic factors.

3. The paper relies heavily on empirical results and intuitive arguments about communication capacity. A more formal information-theoretic analysis would better support these claims. While the perspective is novel, the underlying components are established techniques rather than a fundamental theoretical breakthrough.

---

> ### Author Rebuttal · Authors · 2026-03-30
>
> We thank the reviewer for the insightful and thought-provoking comments. We address the main concerns below.
>
> **(C1) Information-theoretic grounding and channel capacity.**
>
> Our work is primarily empirical, but it does introduce an explicit information-theoretic constraint that was not sufficiently highlighted in the paper.
>
> By construction, the teacher–student interaction is mediated through binary targets of dimension $B$, i.e., $Z \in {0,1}^B$. This implies that the communicated message has bounded entropy: $H(Z) \leq B \ \text{bits}$, and therefore, the mutual information is also bounded: $I(Z_T; Z_S) \leq B$.
>
> This enforces a fixed-capacity communication channel between teacher and student. Importantly, this constraint applies to the communication channel, not to the learned representations themselves, which remain continuous.
>
> Our coding-rate regularization can be interpreted as encouraging efficient use of this bounded capacity by promoting high entropy and low redundancy across dimensions. However, we agree that this remains a surrogate and does not provide a formal guarantee of optimal channel utilization. In particular, coding-rate regularization captures second-order dependencies and encourages non-redundant representations, but does not directly optimize the full joint entropy of the code.
>
> Our use of coding-rate regularization should be understood primarily as a practical and effective choice rather than a central component of the contribution. We do not claim that it provides an optimal or unique approximation of entropy maximization. Other alternatives could be considered to approximate entropy maximization, such as combining bit-wise entropy maximization with decorrelation constraints (e.g., Gram matrix diagonalization), or other objectives explicitly targeting independence across dimensions. We chose coding rate because it has been shown to be effective in prior work (SimDINO) and was empirically robust in our setting as well.
>
> More generally, we view this work as a first step toward understanding how to control information flow in self-supervised learning through discrete communication, and we are currently investigating more principled formulations of this objective.
>
> We will revise the paper to make this explicit and clarify the scope of our information-theoretic interpretation.
>
> **(C2) Discretization at the output vs. earlier layers.**
>
> Exploring discretization at earlier stages or hierarchical discrete representations is an interesting direction for future work. In this work, we deliberately apply the discrete bottleneck only at the output of the projection head, so that it acts as a communication constraint on the supervision signal, rather than on the representation itself. The backbone therefore remains continuous, which preserves its expressivity and facilitates optimization.
>
> Applying discretization earlier in the network would impose much stronger constraints and is known to make optimization significantly more difficult. In particular, training binary neural networks involves discrete optimization and discontinuous activations, which complicates gradient-based learning and typically requires specialized techniques (e.g., straight-through estimators) (Meng et al., 2020; Qin et al., 2020).
>
> In contrast, our formulation keeps the optimization in a continuous space while constraining only the information exchanged between teacher and student. Empirically, this appears sufficient to induce structured representations, as the discrete constraint is propagated back through the network via the training objective.
>
> More broadly, this design can be interpreted as separating the representation space (continuous) from the communication channel (discrete): the model learns rich continuous features, but is trained to express them through a limited discrete interface.
>
> **(C3) Alignment with continuous downstream tasks (e.g., VQA, captioning).**
>
> Our method does not produce discrete representations at inference time. The discrete bottleneck is used only as a training-time supervision mechanism, while the backbone outputs remain continuous.
>
> As a result, the learned representations can still be directly aligned with continuous modalities such as text embeddings. This differs from approaches that explicitly quantize the representation space: we constrain the communication between teacher and student, but not the representation itself.
>
> While we do not evaluate multimodal tasks in this work, our results across downstream tasks suggest that the learned representations retain fine-grained semantic information. Extending this framework to multimodal settings is an interesting direction for future work.

---

> > ### Author Rebuttal · Reviewer_JtoP · 2026-04-03
> >
> > We thank the authors for their clear and well-structured rebuttal. The three main concerns have been satisfactorily addressed.

---

### Official Review · Reviewer_XEME · 2026-03-11

**Soundness:** 3
**Presentation:** 3
**Significance:** 2
**Originality:** 3
**Overall Recommendation:** 4
**Confidence:** 5

**Summary:**

This paper presents a new self-supervised loss for training image representations using a joint-embedding architecture (JEA) that is trained to optimize representation invariances to data augmentation. Assuming that the encoder (and projector) output discrete codes (N binary variables following a Bernoulli distribution), the authors propose training the JEA by maximizing the mutual information I(Z1, Z2) between the codes Z1, Z2 corresponding to each view.

Maximizing the entropy forces the system to avoid collapse solutions, while minimizing the conditional entropy promotes invariance between codes for a given example.

The authors validate their approach by pretraining a ViT-B/16 architecture on the IN1K dataset and report performance for linear classification, k-NN classification, retrieval, as well as downstream task performance on a wide range of tasks (segmentation, fine-grained classification, ...). The results support the claim that the proposed approach outperforms existing SSL methods such as DINO and SimDINO in a controlled setting. The authors also perform an in-depth ablation analysis of the different algorithm components.

**Compliance With Llm Reviewing Policy:**

Affirmed.

**Final Justification:**

The additional experiments comparing with DINOv2 address my most important concern. Overall, the paper is clear and well motivated. The proposed SSL loss is theoretically grounded and provides a principled way to avoid collapse.

**Key Questions For Authors:**

- Given that the loss a provide a principle way to avoid collapse is it necessary to use EMA on the teacher weight ? Additionally, one could remove stop-gradient and backprop through the teacher weight using straight estimator for instance.

- Instead of using coding-rate regularization, one could estimate the empirical entropy on the mini-batch, and use a straight-through estimator to backpropagate through the discrete unit. Did you consider such an approach?

**Limitations:**

Paper does not have a limitation approach. It would be nice to add one discussing the limitation of the current experimental protocol focusing on the VIT.B/16 scale.

**Strengths And Weaknesses:**

Strengths:
- The paper is clear and well motivated. The proposed SSL loss is theoretically grounded and provides a principled way to avoid collapse.
- The authors report performance on an extended set of downstream tasks, showcasing the generality of the representations learned with their SSL approach.
- The results support the claim that the approach outperforms DINO and SimDINO on a ViT-B/16 architecture when controlling for dataset size.
- The authors perform ablation studies to showcase the importance of different loss components.

Weaknesses:
- The main limitation of the paper is the lack of scaling experiments. It's unclear whether the gains obtained from the loss would transfer to ViT-L or ViT-g architectures. Also, by limiting their empirical exploration to ViT-B/16, the reported results are not competitive with larger-scale models such as DINOv2/v3. While running at larger scale is expensive and may not be easy, it would be important to acknowledge this limitation and report the state of the art on each task.
- Additionally, it would be informative to add other SSL baselines as comparisons, such as iBOT, DINOv2, VICReg, and LeJEPA. While I understand that some of these approaches use a combination of local and global losses, it would still be informative for understanding which combinations of losses lead to the best performance from a practitioner perspective.

---

> ### Author Rebuttal · Authors · 2026-03-30
>
> We thank the reviewer for the constructive feedback. We address the main concerns below.
>
> **(C1) Comparison with stronger models.**
>
> Evaluating at larger scale is indeed important. Our goal in this work was to isolate the effect of the proposed objective in a controlled and widely used setting (ViT-B/16, SimDINO protocol), enabling a direct comparison of training objectives without confounding factors related to scale or architecture.
>
> We conducted additional experiments since submission, including comparisons with DINOv2 under the same evaluation protocol. We observe that:
> - BITS-fixed already outperforms DINOv2 (reproduced in our setup) in retrieval (mAP) on ImageNet-1K (43.44 vs 38.33), while being slightly lower in kNN (73.32 vs 75.93) and Linear Probing (76.7 vs 76.8).
> - BITS-reset further improves both retrieval (50.64 vs. 38.33) and linear probing (77.8 vs. 76.8), while remaining slightly lower in k-NN classification (73.5 vs. 75.93).
> We will report these results in the revised version. We will also explicitly acknowledge the limitation of evaluating only at the ViT-B/16 scale.
>
> **(C2) More complex SSL objectives (iBOT, DINOv2, etc.)**
>
> Several of the suggested methods combine multiple objectives. In particular, DINOv2 integrates a patch-level objective inspired by iBOT together with a global alignment loss, and therefore partially covers this class of hybrid approaches.
>
> To further address this point, we implemented a hybrid variant combining our discrete communication framework with an iBOT-style patch-level objective, using a binary cross-entropy formulation with multi-bit discrete targets (8 bits per patch). In this setting, we observe consistent improvements over DINOv2 across all metrics (mAP: 47.03 vs 38.33, k-NN: 76.66 vs 75.93, linear probing: 78 vs 76.8).
>
> This indicates that discrete communication is not only competitive as a standalone objective, but also complementary to local patch-level losses.
>
> **(C3) Role of EMA and teacher–student asymmetry.**
>
> While coding-rate regularization helps mitigate collapse, the teacher network plays an additional role by providing stable and slowly evolving targets. EMA ensures smoother target dynamics, which stabilizes training beyond collapse prevention, as observed in prior self-distillation methods such as BYOL and DINO.
>
> Removing the stop-gradient and backpropagating through the teacher would eliminate the asymmetry between teacher and student, which is known to be critical for stable optimization in non-contrastive SSL. In particular, BYOL relies on this asymmetry through the combination of EMA and stop-gradient (Grill et al., 2020), and DINO also shows that EMA is essential for stable training (Caron et al., 2021, Fig. 6).
>
> In preliminary experiments, we explored a symmetric formulation similar to Barlow Twins, but consistently observed lower performance than with the distillation-based setup.
>
> **(C4) Coding-rate vs. entropy estimation.**
>
> Estimating entropy from mini-batches corresponds to the sum of marginal entropies across dimensions, implicitly assuming independence between bits. While this encourages balanced activations (maximal entropy obtained at p=0.5), it does not prevent redundancy across dimensions, as multiple bits can encode the same information.
>
> In contrast, the relevant quantity is the joint entropy of the full code, which is intractable to estimate in high dimensions. Coding-rate regularization provides a tractable surrogate that captures second-order dependencies and encourages globally diverse and non-redundant representations without explicitly modeling the full joint distribution. Empirically, the effectiveness of this regularization is supported by the high joint entropy observed in Section 5.5: contiguous 8-bit blocks achieve approximately 7.1 bits out of a maximum of 8, indicating that the learned codes are not merely marginally balanced but jointly diverse, which is a property that marginal entropy estimation alone cannot guarantee.
>
> **(C5) Limitations.**
>
> We will add a dedicated discussion of limitations, including the focus on ViT-B/16 and the absence of large-scale experiments.

---

> > ### Author Rebuttal · Reviewer_XEME · 2026-04-01
> >
> > Thank you for your rebuttal. The additional experiments comparing with DINOv2 address my most important concern. Additionally, I would encourage the author to add a discussion on the importance of EMA and coding-rate vs entropy estimation in the paper.

---

### Official Review · Reviewer_8QJD · 2026-03-12

**Soundness:** 3
**Presentation:** 2
**Significance:** 2
**Originality:** 2
**Overall Recommendation:** 4
**Confidence:** 3

**Summary:**

The paper proposes a new self-supervised objective that aligns teacher-student representations independently across dimensions (unlike joint objectives like cosine similarity) and uses discretized teacher embeddings as targets. The paper argues that this objective leads to capturing more fine-grained representations.

**Compliance With Llm Reviewing Policy:**

Affirmed.

**Final Justification:**

I recognize that I might not fully appreciate the advantages of the factorized 0/1 discrete training framework in comparison to other SSL methods. The authors' discussion is helpful, but I strongly encourage them to improve this aspect of the paper and discuss it more clearly.

**Key Questions For Authors:**

- What is the key supporting evidence that discrete targets result in superior performance? Are they strong enough?
- Is it possible to extend BITS to a more performant DINOv2 setting?
- How do authors explain performance degradation under fine-tuning for (Sim)DINO in Tab. 8, but not BITS?

**Limitations:**

The paper does not discuss the limitations of the proposed method and evaluations. For example, it is unclear how this approach would extend to more performant settings, such as DINOv2, and overall marginal gains over baselines, except for Tab. 8.

**Strengths And Weaknesses:**

**Strengths:**
- The idea of using a potentially more expressive learning signal via distributed representations, rather than the classification-like clustering objective of DINO, appears sound and is worth exploring.

**Weaknesses:**
- The paper presents and motivates the method in a rather confusing manner, making it hard to grasp the core components. Specifically, it often mixes the two techniques of 1) discrete targets and 2) objective factorization over dimensions. Having a schematic illustration that clearly distinguishes between prior methods and the proposed method would make it much easier.
- The experimental support for adopting the method is rather limited. Except for fine-grained classification, the improvements seem minor (IN1K, Davis) or rather low overall (COCO unsupervised semseg). In addition, DINO and SimDINO baselines appear underperforming (e.g., Tab. 1 vs. Tab. 2 in DINO), further reducing the gap. Similarly, the DINO-based MaskCut baseline seems to get much better performance in the original paper (~15 AP_50 in their Tab. 6)
- Some of the design choices are not properly studied and ablated.
	- Discrete vs. continuous choice is not properly ablated. In fact, the results in Tab. 10 in the appendix suggest that soft labels achieve rather similar performance, which does not align with the overall claim that discrete labels are crucial. Further investigations would be needed to support this claim. Also, if discreteness is important, how would DINO perform under discrete (but global) targets?
	- The head resetting mechanism isn't properly studied in the (Sim)DINO settings. It remains unclear why it would diverge in those settings but not in BITS. Supporting results and discussion of the effort made to make it work would make it more trustworthy.
- The paper does not propose a way of extending their approach to a more performant setting of DINOv2. Moreover, it suggests incompatibility (L251), which can be a serious limiting factor in scaling this approach further.

More minor:
- The paper refers to "structured" representations in multiple places. Could the authors clarify what exactly they mean by this? How does BITS induce "more" structure compared to the DINO one?

---

> ### Author Rebuttal · Authors · 2026-03-30
>
> We thank the reviewer for the feedback. We address the main concerns below.
>
> **(C1) Clarity of the method and separation of components.**
>
> We believe our method is conceptually grounded in a single principle, namely interpreting self-supervised learning as a discrete communication process between teacher and student. The use of binary targets defines a discrete communication channel with finite capacity (at most d bits), while the learned representations themselves remain continuous.
>
> Factorization across dimensions is not an independent design choice, but a direct consequence of this constraint. Because the channel capacity is limited, redundant or entangled representations become suboptimal, as they waste part of the available bits. The model is therefore encouraged to use each bit efficiently and distribute complementary information across dimensions, leading to a factorized representation. In this sense, factorization emerges from discretization rather than being explicitly imposed.
>
> **(C2) Experimental support and baselines.**
>
> Our method achieves substantial improvements in retrieval, which is a central evaluation in this work. On ImageNet-1K (Tab. 1), BITS-reset improves mAP from 35.68 (DINO) and 38.62 (SimDINO) to 50.64 (+12 pts), with consistent gains under domain shift (Tab. 5). Retrieval directly reflects embedding geometry and therefore provides a strong signal of improved representation structure.
> Classification gains are more moderate, which is expected on a saturated benchmark dominated by global invariances. In contrast, our method targets multi-label structure and factorization, better captured by retrieval and transfer. DAVIS and COCO rely on frozen features under domain shift, where performance differences are typically compressed.
>
> Regarding MaskCut, DINO in our setting is trained under the SimDINO protocol (100 vs. 800 epochs), ensuring fair comparison across methods. MaskCut differences arise because we evaluate frozen features only, without the full CutLER pipeline, following SimDINO evaluation.
>
> **(C3) Discrete vs. continuous targets.**
>
> Tab. 10 actually supports, rather than contradicts, the importance of discretization: it shows that performance improves as targets become more discrete. Soft targets are only competitive at very low temperatures, where they are nearly binary (≈0.99/0.01), while higher temperatures degrade performance or destabilize training.
>
> A similar trend is observed in DINO’s original paper, where sharper targets improve performance. However, fully hard (one-hot) targets degrade performance, as they collapse the categorical channel (K=65,536) to a single label, reducing its capacity to only $\log_2 K \approx 16$ bits.
>
> In contrast, our formulation uses a multi-label binary channel (with 256 bits of information), where information is distributed across dimensions, making hard targets effective. Our results show that this leads to a better trade-off between expressivity and factorization.
>
> **(C4) Head reset in (Sim)DINO.**
>
> Applying head reset in SimDINO consistently leads to divergence (NaNs). More precisely, immediately after the reset, the average cosine similarity jumps from about 0.6 to values very close to 1. This can be explained by the fact that the teacher and student backbones remain highly similar, while the reset head acts as a random projection. In high dimensions, this preserves strong correlations, leading to near-perfect alignment (cosine ≈ 1), which saturates the objective and removes useful learning signal. In contrast, our BCE objective operates at the bit level with bounded gradients and does not rely on global vector alignment, making it robust to such perturbations. This matches our empirical observations: head resetting is unstable in SimDINO but remains stable and beneficial in BITS (Tab. 3).
>
> **(C5) Extension to DINOv2.**
>
> We conducted additional experiments:
> - BITS-fixed already outperforms DINOv2 (reproduced in our setup) in retrieval (mAP) on ImageNet-1K (43.44 vs 38.33), while being slightly lower in kNN (73.32 vs 75.93) and Linear Probing (76.7 vs 76.8).
> - BITS-reset further improves both retrieval (50.64 vs. 38.33) and linear probing (77.8 vs. 76.8), while remaining slightly lower in k-NN classification (73.5 vs. 75.93).
> - A hybrid BITS + iBOT-style variant (8-bit BCE per patch) improves over DINOv2 on all metrics (mAP: 47.03 vs. 38.33, k-NN: 76.66 vs. 75.93, linear probing: 78 vs. 76.8)
>
> **(C7) Fine-tuning behavior.**
>
> A known weakness of classical SSL methods is that they do not encourage decomposition of semantic factors (Locatello et al., 2019). In contrast, discrete communication enforces an explicit allocation of complementary information across dimensions. Empirically, continued training degrades (Sim)DINO but improves BITS (Tab. 8), suggesting better adaptability under domain shift.

---

> > ### Author Rebuttal · Reviewer_8QJD · 2026-04-02
> >
> > **Re C1:**
> > I would like to note that i am not necessarily questioning the potential benefit of the overall proposed framework, but point out that, in my opinion, it is **not presented clearly** and its **relationship to other contrastive SSL frameworks is not clearly established**. Specifically, it is still unclear from the paper and response why the factorization is a "direct consequence of this constraint". In L142 and L150, the paper just defines the communication channel to be the Bernoulli random variable. This choice does not *explicitly* follow from any constraint. As the authors mention in their (C3) response, DINO with hard labels can be seen as a non-factorized discrete categorical communication channel, just with 16 times fewer bits due to the specific choice of the vocabulary size K. In fact, even a continuous vector is represented as a float32 and has a communication capacity of 32 bits, so an 8-dimensional float32 vector would also have a capacity of 256 bits. These are examples of other design choices that are not clearly discussed.
> >
> > **Re C2:**
> > > substantial improvements in retrieval, which is a central evaluation in this work
> >
> > In this case, could the authors clarify why no other retrieval-specific baseline is reported?
> >
> > >DAVIS and COCO rely on frozen features under domain shift, where performance differences are typically compressed.
> >
> > Did authors also try fine-tuning? To clarify, do the authors argue that the presented differences (e.g., AP 2.2 vs 2.8, and J&F 61.83 vs. 62.78) are meaningful? I strongly encourage the authors to include qualitative results for the detection and segmentation tasks, especially on COCO as the absolute numbers appear extremely low.
> >
> > **Re C3:**
> > The paper does not report the distribution of the soft label values to make a conclusion, only the temperature. All the tested values except $\tau=1$ appear to work reasonably well (43.06-43.28 range), and no effect of the hard thresholding on the hard labels is shown to claim the optimal choice for it being less narrow.
> >
> > **Re C4-C5:** Thank you for the additional results and clarifications.
> >
> > **Re C7:**
> > Thanks for the answer. To the best of my understanding, (Locatello et al., 2019) argues that w/o any form of semantic supervision, one should not assume a decomposition of semantic factors. In the case of contrastive learning, this supervision comes in the form of augmentation choice, which is equal between (Sim)DINO and BITS. But the "discrete communication" is a bottleneck similar to bottlenecks in VAEs, which does not explicitly say *what and how* information should be preserved and factorized. Overall, it is rather unclear how this factorization relates to frozen vs. fine-tuning results. Could the authors comment on this?

---

> > > ### Author Response · Authors · 2026-04-07
> > >
> > > We thank the reviewer for the interesting discussion and provide the response elements below.
> > >
> > > **(C1) Factorization and relation to contrastive approaches.**
> > >
> > > Following FroSSL, we can distinguish between sample-level contrastive methods (e.g., SimCLR, MoCo), which rely on instance discrimination, and dimension-level objectives (e.g., Barlow Twins, VICReg, coding-rate), which structure representations across feature dimensions. Our approach belongs to the latter family and builds on non-contrastive self-distillation (e.g., DINO), which has been shown to outperform sample-level contrastive methods in similar settings.
> > >
> > > Our contribution is to introduce an explicit communication constraint in this regime. We model the teacher–student interaction as a discrete channel with finite capacity. This should not be confused with numerical precision: while a float32 variable can represent up to $2^{32}$ values, its entropy depends on the distribution (e.g., $H=32$ bits if uniform, but $H=1$ bit if restricted to $\{0,1\}$). In contrast, our formulation defines a discrete code $Z \in \{0,1\}^B$ with $H(Z)\leq B$, where each dimension contributes independently. Under this constraint, redundancy directly wastes capacity, which induces a pressure to distribute information across dimensions. This is the origin of the observed factorization.
> > >
> > > Finally, following the reviewer’s suggestion, we evaluated a SimCLR objective within our setup. It yields significantly worse retrieval (mAP 25.02 vs. 43.44 for BITS and 50.64 for BITS-reset) and slightly lower kNN (69.52 vs. 73.32 with BITS and 73.5 with BITS-reset ). This is consistent with recent analyses showing that contrastive learning prioritizes alignment, while uniformity, which is critical for retrieval, emerges more slowly.
> > >
> > > **(C2) Retrieval baselines and downstream evaluation.**
> > >
> > > Contrastive methods such as SimCLR are often considered strong retrieval baselines. Following the reviewer’s suggestion (see C1), we evaluated SimCLR in our setup and obtained significantly lower retrieval performance (mAP 25.02 vs. 43.44 for BITS and 50.64 for BITS-reset), confirming that our comparisons cover this regime.
> > >
> > > Regarding DAVIS and COCO, we report frozen-feature evaluation to directly assess representation quality under domain shift. While absolute numbers are low in this setting, relative differences are consistent across datasets and are standard in this evaluation protocol. Fine-tuning typically amplifies these differences but introduces additional factors, making comparisons less controlled. Our goal here is to isolate representation quality rather than optimize task-specific performance.
> > >
> > > **(C3) Soft vs. hard targets.**
> > >
> > > We agree that reporting only temperature is indirect, and we therefore analyze the actual target distributions. For instance, at $\tau=0.1$, 48.33\% of values are $>0.99$ and 48.25\% are $<0.01$; at $\tau=0.01$, this increases to 49.66\% and 49.99\%, respectively. This shows that the stable regimes in Tab.~10 already corresponds to nearly binary targets. We will add those statistics in the appendix.
> > >
> > > The comparison with DINO is also informative. Their paper shows that fully hard (one-hot) targets degrade performance, as they collapse the categorical channel (with $K=65{,}536$) to a single label, reducing its capacity to $\log_2 K \approx 16$ bits. In contrast, our multi-label binary channel preserves capacity across dimensions, which explains why hard targets remain effective in our setting.
> > >
> > > **(C7) Relation to factorization and adaptation.**
> > >
> > > We agree that discretization alone does not guarantee semantic disentanglement (as discussed in Locatello et al.). Our claim is more modest: the discrete, capacity-constrained channel introduces an inductive bias toward distributing information across dimensions.
> > >
> > > This differs from standard bottlenecks such as VAEs, where latent variables remain continuous and can freely mix factors. In contrast, our binary codes impose a stronger structural constraint: each dimension is limited to $\{0,1\}$, and coding-rate regularization encourages efficient use of each bit. This does not enforce semantic factors, but promotes a more explicit allocation of information across dimensions.
> > >
> > > We hypothesize that this structured allocation explains the behavior under domain shift. When representations are more factorized, changes affect only a subset of dimensions, making adaptation easier. In contrast, entangled representations mix factors, so even small shifts can impact the representation globally. This is consistent with the fine-tuning and frozen results observed in Section 5.4.

---

### Official Review · Reviewer_yyA1 · 2026-03-12

**Soundness:** 4
**Presentation:** 4
**Significance:** 4
**Originality:** 4
**Overall Recommendation:** 4
**Confidence:** 5

**Summary:**

The authors propose a novel approach called BITS (Binary Information Transmission for Self-Supervision) that looks at visual SSL as a discrete communication task between teacher and student network. They achieve this through element-wise binary cross entropy loss, a coding-rate regularization technique, and periodic projection head reinitialization. Their method is competitive with prior continous agreement methods for downstream tasks like classification, retrieval, dense prediction (detection/segmentation), and self-supervised domain adaptation.

**Compliance With Llm Reviewing Policy:**

Affirmed.

**Final Justification:**

The authors addressed my concerns during the rebuttal, but C3 was discussed at a high-level. Some concrete experiments would have resulted in an updated score. Given the current scenario, I will be maintaining my original score.

**Key Questions For Authors:**

Please see weaknesses.

**Limitations:**

Yes

**Strengths And Weaknesses:**

# Strengths
- Novel framework: This paper offers an original perspective of SSL by enforcing discrete agreement between teacher and student networks. It replaces continuous alignment with multi-label discrete alignment where the teacher network passes a discrete message to the student network. This also provides them higher degree of explicit control over representational structure and capacity without discretizing the backbone embeddings themselves.
- Improved representation geometry: Fig 1. shows interesting variance analysis where BITS-reset outperforms other methods in terms of expressivity by better information distribution across dimensions. The authors further strengthen this argument by introducing $d_{eff}$ and $r_{eff}$ in Tab 2.
- Comprehensive empirical results:
    - BITS outpaces DINO and SimDINO on ImageNet-1K, with particularly large gains in image retrieval, indicating excellent fine-grained similarity structure
    - Strong classification and retrieval results for domain adaptation, as well as dense tasks (object segmentation/detection and video segmentation)
- Insightful analysis of discrete (semantic) code: The average entropy per bit is ~0.9. Additionally, Fig 3. shows that a single bit does not encode class-level but contextual information. I find these results quite impressive.
- Presentation: The paper is well-written and easy to follow.

# Weaknesses
- While it is throughout highlighted that have discrete embeddings gives more control and improves representation geometry, the evaluations are done mainly on continuous embeddings? I was expecting that the discrete embeddings would also be tested (even if it performs worse)
- In L433-435, “Using the full 256bit representation yields an mAP of 47.59, compared to 50.64 when using the continuous backbone embeddings.”: experimental details are not clarified.
- Suggestions for improving Section 5.2 and 5.4:
    - Section 5.2:  The claim of increased expressivity relies on global effective dimensionality (d_{eff}). To strengthen this claim, the authors should consider decomposing this variance into *directional variation between class centers*, as recently analyzed in [1]. Since [1] show that directional variation dominates linear probing bounds, analyzing these specific components would definitively prove whether the higher (d_{eff}) in BITS-reset actually stems from useful semantic separation rather than just inflated intra-class noise.
    - Section 5.4: The 'robustness-factorization' trade-off observed under severe domain shift in Section 5.4 is empirically interesting but lacks geometric explanation.  The authors could leverage the theoretical framework [1], which bounds linear-probing error based on the interplay between within-class dispersion (global CDNV) and directional variation (directional CDNV). By measuring these two metrics on the out-of-distribution features, the authors could precisely pinpoint why BITS-reset fails in frozen settings!

### References

[1] **Luthra, A., Yang, T., & Galanti, T. (2025).** *Self-Supervised Contrastive Learning Is Approximately Supervised Contrastive Learning.* NeurIPS.

---

> ### Author Rebuttal · Authors · 2026-03-30
>
> We thank the reviewer for the insightful and constructive feedback. We address the main concerns below.
>
> **(C1) Discrete vs. continuous embeddings and binary embeddings evaluation.**
>
> We thank the reviewer for this important point. We would first like to clarify a key aspect of our approach. While we introduce a discrete bottleneck, this applies only to the communication between teacher and student, not to the final learned representations. The backbone outputs remain continuous, and all downstream evaluations are therefore performed on continuous embeddings. In this sense, our method constrains the supervision signal rather than discretizing the representation itself.
>
> Nevertheless, we did also evaluate the discrete binary codes, both quantitatively and qualitatively, to better understand the type of information conveyed through the discrete communication channel and how it relates to the structure of the learned representations. As noted in the paper (L433–435), using the full 256-bit representation of BITS-reset yields an mAP of 47.59, compared to 50.64 when using the continuous backbone embeddings. The retrieval evaluation on binary codes was performed using asymmetric Hamming distance (i.e., comparing binary codes to probabilities), which is standard in hashing-based retrieval. We agree that the experimental details of this experiment should be clarified and will revise the paper accordingly. We also evaluated pure Hamming distance between binary codes, which yields an mAP of 46.86. We will add this result for completeness.
>
> Beyond quantitative performance, we chose to emphasize qualitative analysis of the binary codes (Sec 5.5), which we believe provides more insight into the semantic information captured by individual bits and their reuse across classes.
>
> **(C2) Effective dimensionality and semantic structure.**
>
> We agree that decomposing the variance into more structured components, such as directional variation between class centers, would provide a more precise understanding of the nature of the observed increase in effective dimensionality. While we do not perform this decomposition in the current work, the improvements observed in downstream tasks, particularly in retrieval and linear probing, suggest that the increased effective dimensionality is associated with meaningful semantic structure rather than intra-class variability. The consistent gains across both in-domain and out-of-distribution benchmarks further support this interpretation.
>
> **(C3) Robustness–factorization trade-off.**
>
> This is an interesting perspective. A more refined geometric decomposition could indeed help characterize the robustness–factorization trade-off observed in Section 5.4, and the framework proposed in [1] is relevant in this regard.
>
> In our case, the behavior of BITS-reset can be interpreted as follows: periodic head reinitialization encourages the backbone to reorganize its features across multiple discrete coding schemes, promoting stronger specialization and factorization. While this improves expressivity and adaptation performance, it can reduce robustness in frozen settings when the learned structure does not align with the target domain. Analyzing such geometric quantities on out-of-distribution features would be a valuable way to further understand this effect.

---

> > ### Author Rebuttal · Reviewer_yyA1 · 2026-04-03
> >
> > Thank you for your rebuttal. My concerns have been resolved. I encourage the authors to include experiments on refined geometric decomposition as it will improve the quality of their work, especially section 5.4.
> >
> > Since there weren't additional (computationally inexpensive) experiments conducted to address C3, I am inclined towards maintaining my original score.

---

### Decision · Program_Chairs · 2026-04-30

**Decision:**

Accept (regular)

**Comment:**

The paper frames SSL as a discrete communication task between teacher and student networks.  The paper achieves competitive results with prior continuous agreement methods.

The paper proposes an original perspective of SSL by enforcing a discrete agreement between the networks.  The representation geometry is improved as well.

The motivation of the proposal could be better.  The experimental support for the method is limited, and some of the design choices are not properly evaluated.

Reviewer yyA1 recommended a weak accept claiming that the authors addressed their concerns during the rebuttal but that some concrete experiments about the robustness-factorization trade-off were missing.

Reviewer 8QJD recommended a weak accept claiming that the discussion could be better.

Reviewer XEME recommended a weak accept after the rebuttal which addressed the main concerns.

Reviewer JtoP recommended a weak accept after the rebuttal claiming that their concerns have been addressed.

Given the competitive results and the enthusiasm of the paper, I am leaning towards an accept recommendation.  While the experiments are not as comprehensive, the reviewers still had a positive view of the paper.